# ML4MILP: A Benchmark Dataset for Machine Learning-based Mixed-Integer Linear Programming

## Abstract

Machine learning (ML)-based approaches for solving mixed integer linear programming (MILP) problems have shown significant potential and are growing in sophistication. Despite this advancement, progress in this field is often hindered by the mixed and unsorted nature of current benchmark datasets, which typically lack carefully categorized collections of homogeneous instances. To bridge this gap, we propose ML4MILP, a new open-source benchmark dataset specifically designed for evaluating ML-based optimization algorithms in the MILP domain. Based on the proposed structure and embedding similarity metrics, we used a novel classification algorithm to carefully categorize the collected and generated instances, resulting in a benchmark dataset encompassing 100,000 instances across more than 70 heterogeneous classes. We demonstrate the utility of ML4MILP through extensive benchmarking against a comprehensive suite of algorithms in the baseline library, consisting of traditional exact solvers and heuristic algorithms, as well as ML-based approaches. Our ML4MILP is open-source and accessible at: https://anonymous.4open.science/r/ML4MILP-6BE0.

## 1 Introduction

Mixed Integer Linear Programming (MILP) problems are pivotal in optimization, impacting sectors such as routing (Kaufman et al., 1998; Heydar et al., 2016; Heisterman & Lengauer, 1991), scheduling (Floudas & Lin, 2005; Zhao et al., 2020; Ku & Beck, 2016), network designing (Luathep et al., 2011; Leitner & Leitner, 2010; Gounaris et al., 2016), and resource allocation (Gertphol et al., 2002; Alfa et al., 2016; Ramlogan & Goulter, 1989).

These problems demand optimization of a linear objective function under constraints with integer and continuous variables (Wolsey, 2007; 2020), furnishing robust solutions to complex combinatorial challenges. Given their significance, developing advanced algorithms for MILP is crucial for efficiently solving real-world problems.

Traditional MILP solving strategies can broadly be categorized into two groups (Zhang et al., 2023): exact solving algorithms based on branch and bound (Yokoyama et al., 2015; Adelgren & Gupte, 2022; He et al., 2014), and approximation algorithms based on heuristic (Fischetti & Lodi, 2010; Triadó-Aymerich et al., 2016; Boujelben et al., 2016). Notable representatives of these categories include pseudo-cost branching (Bénichou et al., 1971; Land & Powell, 1979), strong branching (Achterberg et al., 2005; Dey et al., 2024) and hybrid branching (Achterberg & Berthold, 2009; Turner et al., 2023) in branch and bound, and techniques like feasibility pump (Fischetti et al., 2005; Bertacco et al., 2007), evolutionary computing (Rothberg, 2007; Luo et al., 2017) and large neighborhood search (Nepomuceno et al., 2023; Hendel, 2022) in heuristic. However, in real-world scenarios, there is often a need to solve homogeneous MILP with similar combinatorial structures. In this context, traditional methods face the challenge of cold-starting (Zhang et al., 2023; Nair et al., 2020b), as they cannot leverage accumulated solving knowledge to expedite the process.

In response, recent advancements have leveraged machine learning to address MILP problems. Inspired by Gasse's bipartite graph representation of MILP (Gasse et al., 2019), researchers are increasingly adopting neural networks, particularly Graph Neural Networks (GNNs) (Scarselli et al., 2008), to improve the efficacy of traditional algorithms. These ML-based methods enhance branch

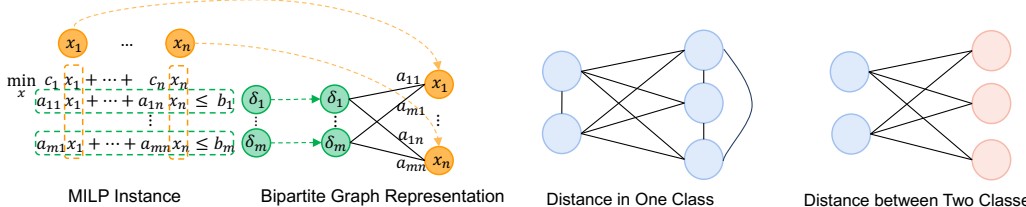

Figure 1: Convert MILP into bipartite graph.   Figure 2: The Calculation of Similarity Score.

decision-making in branch-and-bound (Gupta et al., 2020; Chen et al., 2023a; Gupta et al., 2022), neighborhood selection (Song et al., 2020; Sonnerat et al., 2021; Wu et al., 2021) and predicting high-quality initial solutions in heuristic (Ding et al., 2020; Ye et al., 2023c;b), all aimed at speeding up the solving process.

Despite the effectiveness of machine learning-based algorithms in many applications, our research into recently proposed ML-based algorithms reveals that, although several approaches claim state-of-the-art performance, they lack comprehensive benchmarking and comparison using a standardized benchmark dataset. This deficit obscures the assessment of which techniques perform best under varying conditions, thereby stalling progress in the field.

To address this gap, we present ML4MILP, a new open-source benchmark dataset for evaluating machine learning algorithms in mixed integer linear programming. We hope that ML4MILP will provide researchers with a convenient means to develop and evaluate their methods. Specifically, we have undertaken the following three tasks: **1) Similarity Evaluation.** We proposed a graph statistics-based structural similarity metric and a self-supervised learning GNN-based embedding similarity metric, further achieving fine classification of MILP instances based on GNN embedding. **2) Benchmark Datasets.** We proposed a new standardized MILP problems dataset tailored for evaluating ML-based algorithms, including 100,000 instances across more than 70 classes and their nearly optimal solution gained by adaptive constraint-partitioned algorithm. **3) Baseline Library.** We comprehensively compared and ranked existing mainstream algorithms based on the MILP problem dataset, including objective function values and gaps. Experimental results indicate that some algorithms may not be as robust as claimed under more extensive testing, demonstrating the value of ML4MILP for advancing the field.

In summary, ML4MILP offers the following advantages:

1. **Novelty**: It provides the Learn4MILP community with a new specially designed benchmarking dataset.

2. **Comprehensive**: It also comprehensively tests existing mainstream algorithms, determining which methods truly excel under specific conditions.

3. **Scalability**: ML4MILP also offers a variety of benchmark problems and numerous integrated benchmark algorithms, with plans for regular maintenance and updates to ensure continuous (scalable) expansion.

## 2 RELATED WORK

### 2.1 MIXED INTEGER LINEAR PROGRAMMING PROBLEM

Mixed Integer Linear Programming (MILP) problem is a significant class within combinatorial optimization problems. Formally, a MILP problem can be represented as follows (Wolsey, 2007; 2020):

$$\min_{x} \; c^T x, \text{subject to } Ax \leq b, l \leq x \leq u, x_i \in \mathbb{Z}, i \in \mathbb{I}, \tag{1}$$

where $x$ represents the decision variables, with dimension denoted by $n \in \mathbb{Z}$, and $l, u, c \in \mathbb{R}^n$ correspond to the lower bounds, upper bounds, and coefficient values of the variables, respectively. The matrix $A \in \mathbb{R}^{m \times n}$ and the vector $b \in \mathbb{R}^m$ define the linear constraints of the problem. The set $\mathbb{I} \subseteq \{1, 2, \ldots, n\}$ denotes the indices of variables that are constrained to be integers.

As shown in Figure 1, based on the formulation of MILP, Gasse's proposed MILP bipartite graph representation (Gasse et al., 2019) achieves a lossless translation of the MILP problem into a graph format, serving as input for the neural embedding network (Nair et al., 2020b). More details about MILP and MILP bipartite graph representation are shown in Appendix D.1.

## 2.2 Machine Learning-based Solving Algorithm

As a pioneering effort in leveraging ML-based optimization algorithms for solving MILP problems, Gasse et al. (2019) introduced a novel lossless graph representation approach utilizing bipartite graphs for MILP. Building upon this foundational work, Nair et al. (2020b) from DeepMind developed the concept of neural diving, where initial solution predictions derived from GNNs are employed to fix a majority of the decision variables. Additionally, numerous studies have focused on enhancing the branch-and-bound method, targeting improvements in variable selection (Sun et al., 2020; Balcan et al., 2024), node selection (Labassi et al., 2022; Scavuzzo et al., 2022), and cutting plane strategies (Li et al., 2024; Balcan et al., 2022).

Building upon previous advancements, Sonnerat et al. (2021) from DeepMind further refined the concept of neural diving by introducing NeuralLNS, which enhances the solutions obtained by training a neural network to select search neighborhoods. Following this development, several scholars have explored using reinforcement learning and imitation learning strategies to learn domain selection policies (Wu et al., 2021; Nair et al., 2020a; Chen & Tian, 2019; Liu et al., 2022). In 2023, Ye et al. (2023c) introduced the GNN&GBDT framework, the current state-of-the-art learning-based solving framework, enhancing prediction and iteration capabilities based on NeuralLNS.

Concurrently, several other studies (Ding et al., 2020; Han et al., 2023; Huang et al., 2023) also have attempted to learn optimal solution predictions for MILP problems using GNNs and further refine these predicted solutions. While these methods position themselves as state-of-the-art, their effectiveness could be further validated through testing on a more unified and comprehensive dataset, specifically tailored for ML approaches in the MILP domain.

## 2.3 Related Benchmark Dataset

MILP is a fundamental tool for modeling combinatorial optimization problems, and ML has increasingly been explored as a means to accelerate MILP solving. However, progress in this area is hindered by the limitations of existing benchmark datasets, which often lack standardization, careful categorization, and diverse distributions of problem instances. For example, MIPLIB (Koch et al., 2011) and Coral (Curtis) provide general MILP problem instances but do not offer structured distributions or standardized test sets tailored for ML-based methods. To address some of these shortcomings, Distributional MIPLIB (Huang et al., 2024) introduced a multi-domain library of MILP instances, focusing on curating problems from real-world and existing sources while classifying them into different hardness levels. Expanding this dataset to include larger-scale problem instances and a more comprehensive suite of baseline algorithms would further enhance its utility for evaluating ML frameworks and conducting systematic benchmarking.

Our work, ML4MILP, builds on and addresses these limitations by providing a broader and more diverse range of MILP instances, sourced from both real-world problems and generated cases. These instances are carefully categorized into heterogeneous classes and varying hardness levels using a novel classification algorithm. Unlike previous datasets, ML4MILP includes ultra-large-scale problem instances with up to millions of decision variables and constraints, enabling the evaluation of ML-based methods on more complex and challenging scenarios. Additionally, ML4MILP offers an extensive library of baseline algorithms, including exact solvers and ML-based approaches, along with detailed benchmarking results. These features make ML4MILP a more structured, scalable, and realistic benchmark for advancing ML-guided MILP research.

## 3 Proposed ML4MILP

We have introduced ML4MILP, a new benchmark dataset specifically designed to test ML-based algorithms for solving MILP problems. ML4MILP consists of three main components: Similarity Evaluation, Benchmark Datasets, and Baseline Library. Based on this structure, we conducted

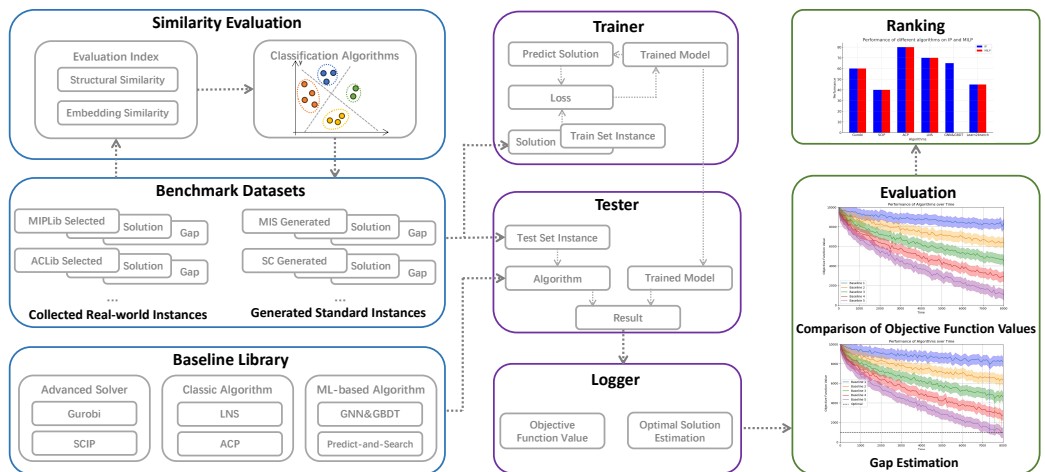

Figure 3: An overview of *ML4MILP* reveals a comprehensive framework. With the extensively collected and carefully designed *Benchmark Datasets*, the evaluation index within the *Similarity Evaluation* component measures the similarity of the collected data, using classification algorithms to recategorize datasets with low similarity scores. This setup facilitates the execution of a "*Trainer-Tester-Logger*" process through the *Baseline Library*, enabling rigorous testing of algorithms and comparisons with classical baselines. Subsequently, the performance of the testing algorithms is evaluated based on objective function values and gap estimates under fixed wall-clock times. Additionally, we score and rank the performance of each baseline under MILP and integer programming (IP) conditions, culminating in a leaderboard of algorithms.

uniform training and testing of baseline algorithms, followed by a comprehensive evaluation and ranking of the results. The framework of ML4MILP is illustrated in Figure3.

## 3.1 SIMILARITY EVALUATION

Since MILP problems can be losslessly encoded as bipartite graphs, Graph Neural Networks (GNNs) have become a common choice for machine learning-based MILP optimization frameworks. Previous studies have demonstrated that machine learning techniques excel in extracting insights from homogeneous MILP datasets (Nair et al., 2020b; Lin et al., 2022), which leads to the fact that the lack of homogeneity poses significant challenges. This issue is not only related to the findings (Oono & Suzuki, 2019), where it was analyzed that GNNs struggle to go deeper due to over-smoothing, resulting in models with insufficient parameters, ultimately affecting generalization in heterogeneous problems; but also to the theoretical analysis (Chen et al., 2023b), which discusses the inadequacy of GNNs in representing general MILPs. We also acknowledge that in recent years, large models with strong out-of-distribution generalization capabilities may have the potential to learn from heterogeneous datasets. However, during the pre-training stage, fine-grained category label distinctions undoubtedly help large models learn the structural differences between different problem categories, which could benefit the development of large model technologies in the optimization community.

Therefore, whether for current machine learning techniques or future new technologies based on large models, fine-grained classified homogeneous datasets are necessary, demonstrating the importance of MILP problem datasets with category labels. However, most collected MILP datasets typically lack homogeneity, which poses significant challenges in identifying and managing dataset uniformity. To overcome these challenges, we introduce innovative *Similarity Evaluation Metrics* and developed a *Classification Algorithm* for re-screening and re-classifying heterogeneous datasets.

### 3.1.1 SIMILARITY EVALUATION METRICS

To improve the assessment of similarity among instances within a dataset, we initially introduced two embedding methods for MILP instances. Specifically, for structure embedding, we detail an embedding approach that represents MILP instances as a 10-dimensional embedding, capturing key

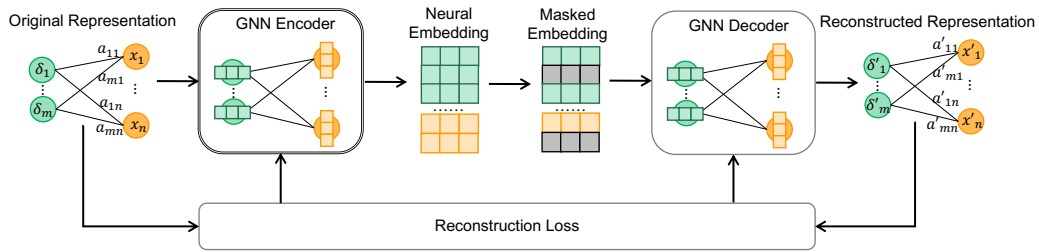

Figure 4: The overview of graph self-supervised learning. For the bipartite graph representation of MILP, we use the encoder-decoder structure for reconstruction, and the loss is computed by comparing the output with the features of the nodes in the original representation.

aspects of the mathematical formulation and bipartite graph characteristics outlined in Section 2.1. This vector includes metrics such as the Fraction of non-zero entries in the coefficient matrix, Mean and standard deviation of the degrees of constraint vertices. The details are shown in Appendix B.2.

However, structural embedding alone may not fully reflect the intricacies introduced by coefficients and local connectivity within problems. To address this, we introduce a neural embedding method using a graph self-supervised learning paradigm, inspired by Graph Autoencoders (Kipf & Welling, 2016). As shown in Figure 4, MILP instances from the MIPLIB Collection are represented as bipartite graphs, which are then encoded using a graph convolutional neural network. During training, parts of this neural embedding are randomly masked, and the model is tasked with reconstructing these missing segments, minimizing the discrepancy between the reconstructed and original graphs.

To quantify the similarity among a set of MILP instances $\mathcal{I}$, we employ the following formula:

$$\text{Embedding Distance} = \frac{\sum_{\mathcal{I}_i, \mathcal{I}_j \in \mathcal{I}, i \neq j} \text{Distance}(\mathcal{I}_i, \mathcal{I}_j)}{|\mathcal{I}|(|\mathcal{I}| - 1)}, \tag{2}$$

where $\text{Distance}(\mathcal{I}_i, \mathcal{I}_j)$ denotes the Euclidean distance between the embeddings of instances $\mathcal{I}_i$ and $\mathcal{I}_j$. As shown in the left of Figure 2, we use the average distance between instance embeddings as a measure to evaluate problem similarity. A lower Embedding Distance indicates a higher degree of homogeneity among the instances, thus affirming their isomorphism within the dataset.

### 3.1.2 CLASSIFICATION ALGORITHM

For datasets that exhibit low similarity, we have developed a novel strategy for re-screening and re-classifying these datasets, thereby retaining only those subsets that demonstrate homogeneity.

Specifically, we consider classifying datasets with low internal similarity into multiple sub-datasets. After graph self-supervised learning shown in Figure 4, we use the GNN encoder to obtain the neural embedding of each problem using the method mentioned in 3.1.1. Then, we employ a spectral clustering algorithm to cluster the neural embeddings within the dataset and recalculate the similarity between the newly formed sub-datasets.

### 3.2 BENCHMARK DATASETS

To accommodate the learning needs of machine learning-based algorithms for solving MILP problems, our dataset comprises two main components: MILP Instances and the corresponding reference Solutions and Gap estimates.

### 3.2.1 MILP INSTANCES

Our MILP instances originate from two primary sources: a new includes real-world scenarios carefully curated and vetted from existing open-source datasets, while the second comprises problems constructed according to standard mathematical formulations.

Firstly, we have carefully gathered a substantial number of MILP instances through various means. These include mainstream open-source, comprehensive datasets such as MIPlib (Koch et al., 2011),

AClib (Hutter et al., 2014), Regions200 (Leyton-Brown et al., 2000), MIRPlib (Papageorgiou et al., 2014), and COR@L (Curtis); domain-specific academic papers, for example, those focusing on the robustness verification of neural networks (Nair et al., 2020b), cut selection (Wang et al., 2023), lot-sizing polytope (Atamtürk & Munoz, 2004), maximizing diffusion in networks (Ahmadizadeh et al., 2010), network designing (Atamtürk, 2002), fixed-charge flow polytope (Atamtürk, 2001), valid inequalities (Atamtürk et al., 2001), conic cuts (Atamtürk & Narayanan, 2010; Şen et al., 2015), and 0-1 knapsack (Atamtürk & Narayanan, 2009); and competitions related to MILP, such as the ML4CO competition in NeurIPS 2021 (Gasse et al., 2022) and the competition on Reoptimization 2023 (Bolusani et al., 2023). The details of each open-source dataset are shown in Appendix A.1.

Secondly, given that the collected problem instances are often small in scale, lacking large-scale examples with millions of decision variables and constraints, we generated a substantial number of standard problem instances based on nine canonical MILP problems: Maximum Independent Set (Tarjan & Trojanowski, 1977), Minimum Vertex Covering (Dinur & Safra, 2005), Set Covering (Caprara et al., 2000), Mixed Integer Knapsack Set (Atamtürk, 2003), Balanced Item Placement (Qu et al., 2022), Combinatorial Auctions (De Vries & Vohra, 2003), Capacitated Facility Location (An et al., 2017). Inspired by Distributional MIPLIB (Huang et al., 2024), we also generated two real-world problem instances: Middle-mile Consolidation Problem with Waiting Times (Greening et al., 2023), and Steiner Network Problem with Coverage Constraints (Huang & Dilkina, 2020). For each type of problem, we generated instances at three levels of difficulty—easy, medium, and hard—corresponding to problem scenarios with tens of thousands, hundreds of thousands, and millions of decision variables, respectively. The details are shown in Appendix A.3.

Upon acquiring the MILP instances, we will utilize the Similarity Evaluation Metrics outlined in Section 3.1.1 to assess the homogeneity of the collected data. For datasets that display significant internal variability, we will apply the Classification Algorithm described in Section 3.1.2 to re-screen and reclassify the problems. This procedure enhances the homogeneity of our benchmarking and facilitates the development of MILP Instances that accurately represent real-world scenarios.

### 3.2.2 SOLUTION AND GAP

For the MILP problem instances, obtaining reference solutions (preferably optimal solutions) and estimating the solution gaps is necessary. To this end, we initially utilize the state-of-the-art solver Gurobi to solve problems for a fixed duration (e.g., 8 hours), during which most problems achieve optimal solutions. These optimal solutions are then packaged into pickle files and the problem instances and saved as part of the ML4MILP training dataset.

However, for some larger-scale problems, particularly the generated standard problems, it is often challenging to obtain optimal solutions within a reasonable time frame. The solutions obtained directly from Gurobi exhibit significant gaps that do not meet the quality requirements for training data reference solutions. Therefore, we introduce the most advanced iterative improvement methods, utilizing an Adaptive Constraints Partition (ACP)-based strategy (Ye et al., 2023a) to iteratively improve the solutions obtained from Gurobi, detailed in Appendix B.1. Then, based on the solution $\overline{x}$ obtained from Gurobi and the associated gap estimate $\overline{g}$, we can compute the gap estimate $g^*$ for the improved solution $x^*$ derived from ACP. Assuming a minimization problem, the gap estimate can be calculated using the following formula $g^* = \frac{x^* - (1-\overline{g})\overline{x}}{x^*}$.

Additionally, we have randomly partitioned the problem data into training and testing datasets. Detailed information about the partitioning scheme and results can be found in Appendix A.4.

### 3.3 BASELINE LIBRARY

To validate the effectiveness of the proposed dataset, we organized the existing mainstream methods into a Baseline Library and conducted comparisons using Benchmark Datasets against these mainstream baselines. The algorithms in the Baseline Library are divided into three parts. a new part includes state-of-the-art large-scale solvers, such as SCIP (version 4.3.0) (Achterberg, 2009) and Gurobi (version 11.0.1) (Gurobi Optimization, 2021), which represent the leading levels of open-source academic and commercial solvers, respectively. We implemented calls to these solvers using their interfaces to solve specific problems. The second part consists of classic solving algorithms, including General Large Neighborhood Search (version ramdom-LNS) (Song et al., 2020) and Adap-

tive Constraint Partition Based Optimization Framework (version ACP2) (Ye et al., 2023a), which we reproduced based on their pseudocode. The third part comprises the latest machine learning-based solving algorithms, including Learn2branch (Gasse et al., 2019), GNN&GBDT-guided framework (Ye et al., 2023c), Neural Diving (Nair et al., 2020b), Predic&Search (Han et al., 2023), Hybrid_Learn2branch (Gupta et al., 2020), and GNN-MILP (Chen et al., 2022). Both algorithms can be trained and tested after adapting them to the problem data.

Furthermore, we also experimented with other solving algorithms such as Feasible Pump (Fischetti et al., 2005) and Simulated Annealing (Abramson & Randall, 1999). However, since these methods failed to obtain feasible solutions within a reasonable computational timeframe for most problems, we ultimately did not include them.

| | Structure Distance | Embedding Distance |
|---|---|---|
| MIS_easy | 0.011 | 1.50e-08 |
| MVC_easy | 0.010 | 1.24e-08 |
| SC_easy | 0.009 | 3.69e-08 |
| Aclib | 25.402 | 2.52e-06 |
| Cut | 7.122 | 1.47e-05 |
| fc.data | 5.867 | 9.68e-07 |
| Hem_knapsack | 0.566 | 8.75e-07 |
| Hem_mis | 0.356 | 4.29e-07 |
| Hem_setcover | 0.286 | 5.33e-07 |
| Hem_corlat | 5.956 | 3.34e-07 |
| Hem_mik | 226.520 | 9.54e-06 |
| item_placement | 0.037 | 7.03e-12 |
| load_balance | 0.436 | 6.97e-07 |
| anonymous | 222.061 | 8.99e-06 |
| nn_verification | 18.460 | 3.71e-06 |
| vary_bounds_s1 | 0.001 | 7.08e-08 |
| vary_bounds_s2 | 0.009 | 3.26e-08 |
| vary_bounds_s3 | 0.009 | 1.90e-08 |
| vary_matrix_s1 | 0.011 | 0.0 |
| vary_matrix_rhs_bounds_s1 | 0.008 | 1.74e-08 |
| vary_obj_s1 | 3.18e-06 | 8.39e-07 |
| vary_obj_s2 | 1.22e-05 | 2.64e-07 |
| vary_rhs_s1 | 0.007 | 9.83e-08 |
| vary_rhs_s2 | 0.550 | 1.11e-05 |
| vary_rhs_s4 | 0.601 | 1.47e-05 |
| Transportation | 148.368 | 1.10e-05 |
| Coral | 1056768.799 | 6.03e-03 |
| ECOGCNN | 468055.052 | 5.41e-02 |
| MIPlib | 7002615758.102 | 8.82e-02 |
| Nexp | 5511575859.716 | 2.48e-01 |

Table 1: The distance of graph structure and neural embedding in each heterogeneous class. The blue background color indicates classical MILP datasets.

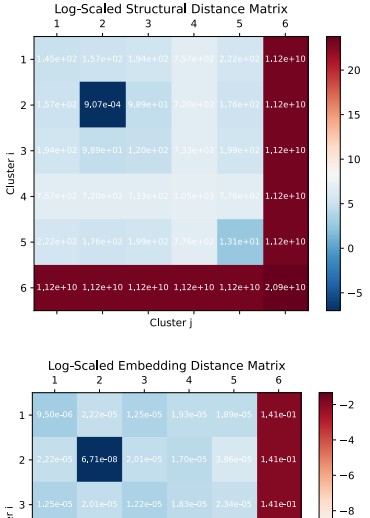

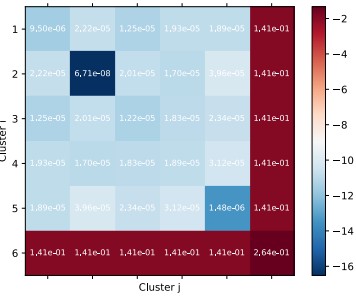

Figure 5: Structure & embedding distance matrix within classes.

# 4 EXPERIMENTS

To validate the effectiveness of ML4MILP, we first conducted a detailed comparison with classic MILP datasets such as MIPLib (Section 4.1), demonstrating the advantage of ML4MILP. Then, we show the distance matrix between categories(Section 4.2), demonstrating the effectiveness of the classification algorithm. Further, based on a selected set of the problem categories from the instance dataset, we conducted comparative tests with algorithms from the Benchmark Library (Section 4.3), indicating that some of the purportedly advanced algorithms did not perform as well as claimed in more comprehensive tests, highlighting ML4MILP's positive role in advancing the field.

## 4.1 DATASET ANALYSIS

To validate the benefits of ML4MILP over traditional MILP datasets such as MIPLib, Coral, and Nexp in evaluating ML-based optimization algorithms, we conducted a comparative analysis of problem similarity within ML4MILP. The settings are shown in Appendix C.2.

The results, detailed in Table 1, reveal a stark contrast in problem similarity. For graph structural embedding, the distances between instances in traditional MILP datasets like MIPLib were found to be several orders of magnitude larger—by hundreds of millions of times—than those in ML4MILP. Similarly, the neural embedding distances in traditional MILP datasets were often tens of thousands

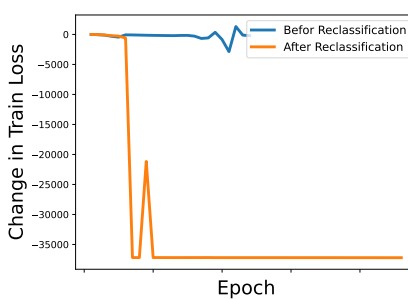 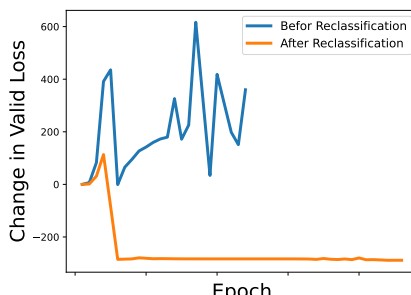

Figure 6: Training loss before and after reclassification.

Figure 7: Validation loss before and after reclassification.

of times greater than those in ML4MILP. These findings demonstrate that ML4MILP provides significantly lower variability between instances, both in terms of problem structure and neural embedding. This confirms ML4MILP's substantial advantages in promoting homogeneity among dataset instances, making it a superior choice for testing ML-based optimization algorithms.

## 4.2 DATASET RECLASSIFICATION

To address the complexities arising from mixed problem instances in datasets, particularly those sourced from open repositories, we employed the spectral clustering algorithm for effective categorization. Taking MIPLib as an illustrative case, we implemented this algorithm, successfully segregating the dataset into six distinct classes. As illustrated on the right side of Figure 2, we utilized the average Euclidean distance to evaluate similarities between these classes. The distances among instances within each class are depicted in Figure 5. Our analysis revealed that the structure and neural embedding distances among a new five classes were notably small, indicating a high degree of similarity. Conversely, the sixth class exhibited significantly larger intra-class and inter-class distances, suggesting the presence of outlier instances. Based on these findings, we opted to exclude this sixth class from the MIPLib dataset to ensure a more homogeneously classified dataset for subsequent baseline testing.

To further substantiate the importance of a homogeneous dataset and the effectiveness of our classification approach, we conducted additional experiments using a semi-convolutional structured GNN (Nair et al., 2020b), a widely recognized neural network. We set the learning rate to $1 \times 10^{-3}$ and divided the dataset into training and validation sets in a 6:4 ratio. Our tests on the MIPLib dataset before and after reclassification, with results displayed in Figures 6 and 7, demonstrated significant differences. The training loss for the original MIPLib, characterized by numerous heterogeneous problems, exhibited slow decreases and considerable fluctuations in validation loss, occasionally leading to NaN (not a number) values. This indicated instability in the GNN training process and a lack of effective learning. In contrast, following reclassification, both training and validation losses decreased steadily, underscoring the critical role of homogeneous datasets in enhancing the performance of machine learning-based MILP optimization frameworks. These findings highlight the necessity of our proposed dataset and reinforce the effectiveness of our classification strategy. Detailed classification results for other datasets, are provided in Appendix C.3.

## 4.3 BENCHMARKING STUDY

To validate the effectiveness of ML4MILP and assess the performance of baseline algorithms across various problem classes, we conducted comparative experiments focused on objective function values and gap estimation under the same wall-clock time limit. Results for six representative baseline methods on benchmark problems are shown in Tables 2 and 3, with full experimental details found in Appendix C.5.

Our findings reveal that, despite many ML-based methods claiming to surpass Gurobi, it consistently outperforms other approaches in most instances, particularly in real-world scenarios like load_balancing and Transportation. This underscores the robustness of classical solvers like Gurobi,

|  | Gurobi | LNS | ACP | Learn2branch | GNN&GBDT | Predict&Search | Time |
|---|---|---|---|---|---|---|---|
| MIS_hard | 2.17e+05 | 2.17e+05 | 2.27e+05 | + | **2.27e+05** | + | 4000s |
| MVC_hard | 2.83e+05 | 2.74e+05 | 2.76e+05 | + | **2.72e+05** | + | 4000s |
| SC_hard | 3.20e+05 | 1.73e+05 | **1.70e+05** | + | 2.29e+05 | + | 4000s |
| MIPlib | **1.84e+04** | 1.98e+04 | 1.84e+04 | 1.89e+04 | - | 1.84e+04 | 150s |
| Coral | **3805.7000** | 4.67e+08 | 1.40e+08 | + | - | 14.5999 | 4000s |
| Cut | **2.89e+04** | 3.35e+04 | 3.07e+04 | 3.71e+04 | - | 2.93e+04 | 4000s |
| ECOGCNN | **7.56e+05** | 7.58e+05 | 7.57e+05 | + | - | **7.56e+05** | 4000s |
| HEM_knapsack | 422.6000 | 422.6000 | 422.6000 | 422.6000 | 422.6000 | 422.6000 | 100s |
| HEM_corlat | 251.0000 | 248.8000 | 251.0000 | ! | - | 251.0000 | 100s |
| HEM_mik | **-6.28e+04** | -6.25e+04 | -6.18e+04 | ! | - | -6.28e+04 | 100s |
| item_placement | **5.3000** | 12.8000 | 10.7000 | 16.5000 | - | 5.5310 | 4000s |
| load_balancing | 708.8000 | 723.2000 | 709.3000 | + | - | 708.8000 | 1000s |
| anonymous | 2.50e+05 | 2.04e+06 | 5.29e+05 | + | - | **2.46e+05** | 4000s |
| Nexp | 1.16e+08 | 1.18e+08 | **1.16e+08** | 1.18e+08 | - | 1.16e+08 | 4000s |
| Transportation | **1.24e+06** | 1.40e+06 | 1.28e+06 | 1.31e+06 | - | 1.25e+06 | 4000s |
| vary_bounds_s1 | **1.24e+04** | 2.07e+04 | **1.24e+04** | 1.29e+04 | - | 1.24e+04 | 400s |
| vary_matrix_s1 | **61.6000** | 61.6000 | 61.6000 | 62.7000 | - | 61.5939 | 100s |
| vary_obj_s1 | 8625.4000 | 8642.0000 | 8625.4000 | 8633.6000 | 8625.4000 | 8625.4000 | 100s |
| vary_rhs_s1 | **-349.5000** | -54.4000 | -291.5000 | + | - | -349.4640 | 100s |
| Aclib | **8.24e+04** | 8.25e+04 | 8.28e+04 | ! | - | 8.24e+04 | 100s |
| fc.data | 378.6000 | 490.4000 | **378.6000** | ! | - | 378.6000 | 100s |
| nn_verification | -8.3000 | -9.7000 | -9.7000 | ! | - | **-8.2514** | 100s |

Table 2: Objective function value of baselines. + represents the problem of scale being too large to accept the time to collect training samples. ! represents the problem of errors during band training. - represents MILP problems that the IP framework, GNN&GBDT, cannot solve.

|  | Gurobi | LNS | ACP | Learn2branch | GNN&GBDT | Predict&Search | Time |
|---|---|---|---|---|---|---|---|
| MIS_hard | 0.1714 | 0.1714 | 0.1184 | + | **0.1169** | + | 4000s |
| MVC_hard | 0.1310 | 0.1018 | 0.1077 | + | **0.0951** | + | 4000s |
| SC_hard | 0.9920 | 0.9852 | **0.9850** | + | 0.9887 | + | 4000s |
| MIPlib | **0.0000** | 0.2157 | 0.0004 | 0.3363 | - | 1.23e-06 | 150s |
| Coral | **2.85e+04** | 3.05e+04 | 3.05e+04 | + | - | 2.33e+04 | 4000s |
| Cut | **0.1490** | 0.2744 | 0.1651 | 0.5782 | - | 0.1568 | 4000s |
| ECOGCNN | **0.2512** | 0.2730 | 0.2516 | + | - | **0.2512** | 4000s |
| HEM_knapsack | **0.0000** | **0.0000** | **0.0000** | **0.0000** | **0.0000** | **0.0000** | 100s |
| HEM_corlat | **0.0000** | 0.0129 | **0.0000** | ! | - | **0.0000** | 100s |
| HEM_mik | **0.0000** | 0.0034 | 0.0146 | ! | - | 3.00e-15 | 100s |
| item_placement | **0.6481** | 0.8431 | 0.8076 | 4.17e+07 | - | 0.6595 | 4000s |
| load_balancing | 0.0028 | 0.0227 | 0.0035 | + | - | **0.0028** | 1000s |
| anonymous | 0.3088 | 0.9256 | 0.5449 | + | - | **0.2909** | 4000s |
| Nexp | 0.0787 | 0.1095 | **0.0754** | 0.1629 | - | 0.0759 | 4000s |
| Transportation | **0.1512** | 0.2490 | 0.1767 | 0.2725 | - | 0.1568 | 4000s |
| vary_bounds_s1 | **0.0000** | 0.3956 | **0.0000** | 0.1518 | - | 0.0003 | 400s |
| vary_matrix_s1 | **0.0000** | 0.0008 | 0.0008 | 0.4002 | - | **0.0000** | 100s |
| vary_obj_s1 | **0.0000** | 0.0019 | **0.0000** | 0.0054 | **0.0000** | **0.0000** | 100s |
| vary_rhs_s1 | 0.0003 | 5.5134 | 2.0037 | + | - | **0.0000** | 100s |
| Aclib | **0.0000** | 0.0006 | 0.0028 | ! | - | **0.0000** | 100s |
| fc.data | **0.0000** | 0.1729 | **0.0000** | ! | - | **0.0000** | 100s |
| nn_verification | 0.0001 | 0.1493 | 0.1493 | ! | - | **0.0000** | 100s |

Table 3: Gap estimation of baselines.

especially when optimality and feasibility guarantees are critical. In contrast, exact ML-based algorithms like learn2branch, despite their theoretical guarantees, struggle with the exponential increase in search space for large-scale problems, as seen in benchmarks like Coral and Cut, where they require significant computational time.

Heuristic-based algorithms, such as GNN&GBDT, excel in large-scale problems like MIS_hard and SC_hard due to effective dimensionality reduction techniques, making them suitable for ultra-large instances. However, GNN&GBDT struggles with certain MILP problems, reflecting the need for more versatile approaches. Predict&Search, which combines machine learning with traditional search techniques, also demonstrates competitive performance, achieving near-optimal results in problems like vary_matrix_s1 and vary_rhs_s1. However, like other heuristic methods, it lacks the guarantees of exact solvers and may fail to find feasible solutions for highly constrained problems.

Our experiments suggest that hybrid strategies—combining heuristics with traditional methods like Predict&Search—can improve results for complex MILP problems by balancing solution quality and efficiency. This highlights the importance of categorized datasets like ML4MILP, which support

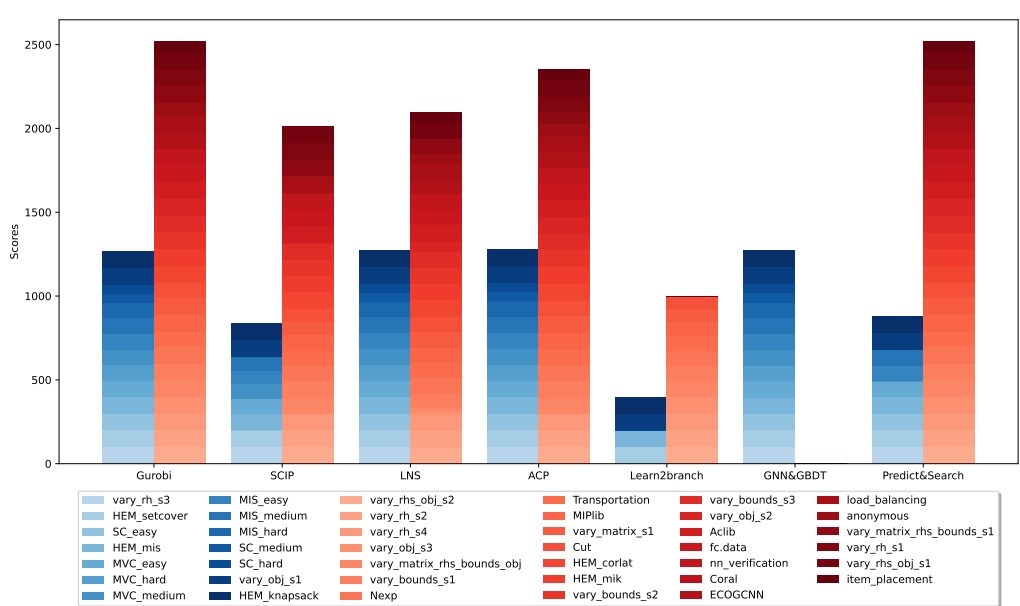

Figure 8: Scores for IP (blue) and MILP (red) problems in ML4MILP for different baselines.

detailed performance analysis and foster the development of more robust hybrid methods leveraging the strengths of existing algorithms.

To enable a direct comparison of baseline algorithms across different problems, we quantified each algorithm's performance by scoring their gap estimates on identical problems. The scores were transformed into a 0–100 scale using a normal distribution function, $\text{Score}(x) \sim \frac{1}{\sqrt{2\pi}\sigma} e^{-\frac{(x-\mu)^2}{2\sigma^2}}$, where $x$ represents the gap estimation. This approach aligns with the central limit theorem, ensuring a more objective performance assessment. Despite the claims of state-of-the-art performance by algorithms like LNS, ACP, Learn2branch, GNN&GBDT, and Predict&Search, the results, depicted in Figure 8, shows that these methods often lack comprehensive evaluation across diverse and large-scale scenarios. For instance, while GNN&GBDT performs exceptionally well in large-scale integer programming problems, other methods like Predict&Search struggle due to higher computational complexity. Conversely, in MILP problems, Predict&Search slightly surpasses Gurobi, yet most other methods fall short of Gurobi's robust performance.

These discrepancies highlight a critical issue: many of the existing machine learning-based algorithms have been evaluated in selective or limited scenarios, failing to provide a clear, holistic view of their true performance across a wide range of problem types and scales. Without a standardized benchmarking and comprehensive comparison, it is difficult to accurately assess which techniques excel under specific conditions. This is where the ML4MILP framework proves invaluable. By offering a unified, rigorous benchmark across various problem domains and scales, ML4MILP enables a much-needed, in-depth comparison of algorithm performance. This framework allows for a clearer understanding of each method's strengths and weaknesses, ultimately driving progress in the field by providing a robust basis for future algorithm development and optimization.

## 5 CONCLUSION AND FUTURE WORK

We propose ML4MILP, a new open-source benchmark dataset designed to evaluate machine learning algorithms in MILP. Through extensive testing, we uncovered significant challenges faced by current ML-based optimization algorithms, highlighting the need for further research to drive advancements in this domain. While ML4MILP has made considerable strides, we recognize areas for improvement. We plan to expand the problem set to include a wider variety of MILP tasks, ensuring richer and more complex challenges. These efforts will ensure ML4MILP remains a crucial tool for advancing ML-based optimization algorithms in MILP, fostering developments in the field.

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

# APPENDIX

This Appendix is divided into four sections. Appendix A provides details of the benchmark dataset. Appendix B outlines the specifics of the algorithms. Appendix C presents additional experimental results to further demonstrate the effectiveness and efficiency of ML4MILP. Finally, Appendix D includes supplementary information on mixed-integer linear programs.

## A    DETAILS OF BENCHMARK DATASET

### A.1    OPEN-SOURCE DATASETS

We have meticulously assembled a substantial collection of mixed integer linear programming (MILP) instances from a variety of sources, including open-source, comprehensive datasets such as MIPlib (Koch et al., 2011), AClib (Hutter et al., 2014), Regions200 (Leyton-Brown et al., 2000), COR@L (Curtis), and MIRPLIB (Papageorgiou et al., 2014); domain-specific academic papers, for example, those focusing on the robustness verification of neural networks (Nair et al., 2020b), cut selection (Wang et al., 2023), lot-sizing polytope (Atamtürk & Munoz, 2004), maximizing diffusion in networks (Ahmadizadeh et al., 2010), network designing (Atamtürk, 2002), fixed-charge flow polytope (Atamtürk, 2001), valid inequalities (Atamtürk et al., 2001), conic cuts (Atamtürk & Narayanan, 2010; Şen et al., 2015), and 0-1 knapsack (Atamtürk & Narayanan, 2009); and competitions related to MILP, such as the ML4CO competition in NeurIPS 2021 (Gasse et al., 2022) and the competition on Reoptimization 2023 (Bolusani et al., 2023). Details such as the number of instances per problem (size of training and testing datasets), the average number of decision variables, and the average number of constraints are all listed in Table 4. Due to double-blind review requirements, the source URLs for the datasets have been excluded and will be made available after the review process is complete.

Our analysis reveals that the ML4MILP dataset is well-populated and robust in terms of instance count and scale. It encompasses a wide range of problem characteristics, including various numbers of decision variables, constraint counts, and densities of coefficient matrices. This diversity makes ML4MILP a broadly comprehensive and extensive dataset suitable for evaluating different aspects of MILP problem-solving through machine learning techniques.

### A.2    USED ASSETS

ML4MILP is an open-sourced tool, and it can be accessed at: Link. Table 5 lists the resources or assets utilized in ML4MILP, along with their respective licenses. It is important to note that we adhere strictly to these licenses during the development of ML4MILP.

"Need to cite" means there is no explicit license, but the repository states that the corresponding article needs to be cited to use the dataset. For datasets prefixed with "vary" in the ML4MILP collection, these originate from the MIP Workshop 2023 Computational Competition (MIPcc23) (Bolusani et al., 2023). It is essential to cite the corresponding literature when using these datasets. Similarly, datasets such as Coral (Linderoth & Ralphs, 2005), MIPlib (Gleixner et al., 2021), and Transportation (Gottlieb & Paulmann, 1998) also require proper citations when utilized. This ensures that all sources are appropriately acknowledged and that the scholarly contributions of these resources are recognized in any analysis or publication that employs them. This practice not only upholds academic integrity but also supports the continuity and openness of research in the field of mixed integer linear programming.

"Readme" means there is no explicit license, but the repository's Readme file explains that it can be used without restriction. For the datasets labeled as Aclib, Cut, fc.data, and Nexp, detailed information and guidelines can be found in the Readme files hosted at the Link. For the ECOGCNN dataset, the Readme file is available at Link.

### A.3    STANDARD PROBLEM INSTANCE

We generated a substantial number of standard problem instances based on nine canonical MILP problems: Maximum Independent Set (MIS) (Tarjan & Trojanowski, 1977), Minimum Vertex Covering (MVC) (Dinur & Safra, 2005), Set Covering (SC) (Caprara et al., 2000), Mixed Integer Knap-

| Name(Path) | Number(Train) | Number(Test) | Avg.Vars | Avg.Constrains |
|---|---|---|---|---|
| nn_verification | 3613 | 9 | 7144.02 | 6533.58 |
| item_placement | 9990 | 10 | 1083 | 195 |
| load_balancing | 9990 | 10 | 61000 | 64307.19 |
| anonymous | 134 | 4 | 34674.03 | 44498.19 |
| HEM_knapsack | 9995 | 5 | 720 | 72 |
| HEM_mis | 9979 | 5 | 500 | 1953.48 |
| HEM_setcover | 9995 | 5 | 1000 | 500 |
| HEM_corlat | 1979 | 5 | 466 | 486.17 |
| HEM_mik | 85 | 5 | 386.67 | 311.67 |
| vary_bounds_s1 | 45 | 5 | 3117 | 1293 |
| vary_bounds_s2 | 45 | 5 | 1758 | 351 |
| vary_bounds_s3 | 45 | 5 | 1758 | 351 |
| vary_matrix_s1 | 45 | 5 | 802 | 531 |
| vary_matrix_rhs_bounds_s1 | 45 | 5 | 27710 | 16288 |
| vary_matrix_rhs_bounds_obj | 45 | 5 | 7973 | 3558 |
| vary_obj_s1 | 45 | 5 | 360 | 55 |
| vary_obj_s2 | 45 | 5 | 745 | 26159 |
| vary_obj_s3 | 45 | 5 | 9599 | 27940 |
| vary_rhs_s1 | 45 | 5 | 12760 | 1501 |
| vary_rhs_s2 | 45 | 5 | 1000 | 1250 |
| vary_rhs_s3 | 45 | 5 | 63009 | 507 |
| vary_rhs_s4 | 45 | 5 | 1000 | 1250 |
| vary_rhs_obj_s1 | 45 | 5 | 90983 | 33438 |
| vary_rhs_obj_s2 | 45 | 5 | 4626 | 8274 |
| Aclib | 89 | 10 | 181 | 180 |
| Coral | 272 | 7 | 18420.92 | 11831.01 |
| Cut | 11 | 3 | 4113 | 1608.57 |
| ECOGCNN | 41 | 3 | 36808.25 | 58768.84 |
| fc.data | 15 | 5 | 571 | 330.5 |
| MIPlib | 46 | 4 | 7719.98 | 6866.04 |
| Nexp | 72 | 5 | 9207.09 | 7977.14 |
| Transportation | 27 | 5 | 4871.5 | 2521.467 |
| MIPLIB_collection_easy | 639 | 10 | 119747.4 | 123628.3 |
| MIPLIB_collection_hard | 102 | 5 | 96181.4 | 101135.8 |
| MIPLIB_collection_open | 199 | 5 | 438355.9 | 258599.5 |
| MIRPLIB_Original | 67 | 5 | 36312.2 | 11485.8 |
| MIRPLIB_Maritime_Group1 | 35 | 5 | 13919.5 | 19329.25 |
| MIRPLIB_Maritime_Group2 | 35 | 5 | 24639.8 | 34053.25 |
| MIRPLIB_Maritime_Group3 | 35 | 5 | 24639.8 | 34057.75 |
| MIRPLIB_Maritime_Group4 | 15 | 5 | 4343.0 | 6336.0 |
| MIRPLIB_Maritime_Group5 | 15 | 5 | 48330.0 | 66812.0 |
| MIRPLIB_Maritime_Group6 | 15 | 5 | 48330.0 | 66815.0 |

Table 4: Detailed parameter information for each open source dataset.

sack Set (MIKS) (Atamtürk, 2003), Balanced Item Placement (BIP) (Qu et al., 2022), Combinatorial Auctions (CA) (De Vries & Vohra, 2003), Capacitated Facility Location (CFL) (An et al., 2017), Middle-mile Consolidation Problem with Waiting Times (MMCW) (Greening et al., 2023), and Steiner Network Problem with Coverage Constraints (SNPCC) (Huang & Dilkina, 2020).

For each type of problem, we generated instances at three levels of difficulty—easy, medium, and hard—corresponding to problem scenarios with tens of thousands, hundreds of thousands, and millions of decision variables, respectively. The details of each problem type and the specific parameters used for generating instances at different difficulty levels are provided below.

**Maximum Independent Set problem**: Consider an undirected graph $\mathcal{G} = (\mathcal{V}, \mathcal{E})$, a subset of nodes $\mathcal{S} \in \mathcal{V}$ is called an independent set iff there is no edge $e \in \mathcal{E}$ between any pair of nodes in $\mathcal{S}$. The maximal independent set problem is to find an independent set in $\mathcal{G}$ of maximum cardinality. If we set binary variable vector $x$ as the decision variables. $x_v = 1$ determines node $v \in \mathcal{V}$ is is chosen in the independent set, and 0 otherwise, MIS problem can be represented as follows.

$$max \sum_{v \in \mathcal{V}} x_v$$
$$s.t. \ x_u + x_v \le 1, \forall (u, v) \in \mathcal{E}, \qquad (3)$$
$$x_v \in \{0, 1\}, \forall v \in \mathcal{V}.$$

**Minimum Vertex Covering problem**: Consider an undirected graph $\mathcal{G} = (\mathcal{V}, \mathcal{E})$, a subset of nodes $\mathcal{S} \in \mathcal{V}$ is called a covering set iff for any edge $e \in \mathcal{E}$ at least one of its endpoints is included in the set $\mathcal{S}$. The minimum vertex covering problem is to find a covering set in $\mathcal{G}$ of minimum cardinality.

| Name(Path) | License | Source |
|---|---|---|
| nn_verification | CC BY 4.0 | Link |
| item_placement | BSD 3-Clause License | Link |
| load_balancing | BSD 3-Clause License | Link |
| anonymous | BSD 3-Clause License | Link |
| HEM_knapsack | MIT | Link |
| HEM_mis | MIT | Link |
| HEM_setcover | MIT | Link |
| HEM_corlat | MIT | Link |
| HEM_mik | MIT | Link |
| vary_bounds_s1 | Need to cite | Link |
| vary_bounds_s2 | Need to cite | Link |
| vary_bounds_s3 | Need to cite | Link |
| vary_matrix_s1 | Need to cite | Link |
| vary_matrix_rhs_bounds_s1 | Need to cite | Link |
| vary_matrix_rhs_bounds_obj | Need to cite | Link |
| vary_obj_s1 | Need to cite | Link |
| vary_obj_s2 | Need to cite | Link |
| vary_obj_s3 | Need to cite | Link |
| vary_rhs_s1 | Need to cite | Link |
| vary_rhs_s2 | Need to cite | Link |
| vary_rhs_s3 | Need to cite | Link |
| vary_rhs_s4 | Need to cite | Link |
| vary_rhs_obj_s1 | Need to cite | Link |
| vary_rhs_obj_s2 | Need to cite | Link |
| Aclib | Readme | Link |
| Coral | Need to cite | Link |
| Cut | Readme | Link |
| ECOGCNN | Readme | Link |
| fc.data | Readme | Link |
| MIPlib_collection_easy | Need to cite | Link |
| MIPlib_collection_hard | Need to cite | Link |
| MIPlib_collection_open | Need to cite | Link |
| MIRPLIB_Original | BSD 3-Clause License | Link |
| MIRPLIB_Maritime_Group1 | BSD 3-Clause License | Link |
| MIRPLIB_Maritime_Group2 | BSD 3-Clause License | Link |
| MIRPLIB_Maritime_Group3 | BSD 3-Clause License | Link |
| MIRPLIB_Maritime_Group4 | BSD 3-Clause License | Link |
| MIRPLIB_Maritime_Group5 | BSD 3-Clause License | Link |
| MIRPLIB_Maritime_Group6 | BSD 3-Clause License | Link |
| Nexp | Readme | Link |
| Transportation | Need to cite | Link |

Table 5: License of each open source dataset. "Need to cite" means there is no explicit license, but the repository states that the corresponding article needs to be cited to use the dataset and it can be used without restriction. "Readme" means there is no explicit license, but the repository's Readme file explains that it can be used without restriction.

If we set binary variable vector $x$ as the decision variables. $x_v = 1$ determines node $v \in \mathcal{V}$ is chosen in the covering set, and 0 otherwise, MVC problem can be represented as follows.

$$
\begin{aligned}
min \sum_{v \in \mathcal{V}} & x_v \\
s.t.\ x_u + x_v &\geq 1, \forall (u, v) \in \mathcal{E}, \\
x_v &\in \{0, 1\}, \forall v \in \mathcal{V}.
\end{aligned}
\tag{4}
$$

**Minimum Vertex Covering problem**: Given a finite set $\mathcal{U} = \{1, 2, \ldots, n\}$ and a collection of $m$ subsets $S_1, \ldots, S_n$ of $\mathcal{U}$, where each subset $S_i$ is associated with a cost $c_i$. The SC problem involves selecting a combination of these subsets such that every element in the universal set $\mathcal{U}$ is included in at least one of the chosen subsets, while minimizing the total cost of the selected subsets. In mathematical terms, we define a binary selection variable $x_i$ for each subset $S_i$, where $x_i = 1$ indicates that the subset $S_i$ is selected and $x_i = 0$ otherwise. SC problem can be represented as follows.

$$
\begin{aligned}
min \sum_{i=1}^{m} & x_i * c_i \\
s.t. \sum_{i=1}^{m} & x_i * (U_j \in S_i) \geq 1, \forall j \in [1, n], \\
x_i &\in \{0, 1\}, \forall i.
\end{aligned}
\tag{5}
$$

**Mixed Integer Knapsack Set Problem**: The Mixed Integer Knapsack Set (MIKS) problem is a variant of resource allocation problems, where a collection of sets is used to cover a set of items, but with the flexibility that some sets can be partially selected while others must be fully included. This problem is commonly encountered in logistics, data center management, and resource allocation, where both discrete and continuous decisions are required. Given $N$ sets and $M$ items, each item must be covered by at least one of the sets. The goal is to minimize the total cost of selecting the sets, where some decision variables are binary (indicating full inclusion or exclusion of a set) and others are continuous (allowing partial inclusion). Let $x_i$ represent the decision variable for set $i$, where $x_i = 1$ if set $i$ is fully selected, and $0 \leq x_i \leq 1$ if set $i$ is partially selected. Each item $j$ must be covered by at least one set that contains it. The problem can be formulated as follows:

$$
\begin{aligned}
\min \quad & \sum_{i=1}^{N} c_i x_i \\
s.t. \quad & \sum_{i:j \in S_i} x_i \geq 1, \quad \forall j \in \{1, 2, \ldots, M\}, \\
& 0 \leq x_i \leq 1, \quad \forall i \in \{1, 2, \ldots, N\}, \\
& x_i \in \{0, 1\} \text{ or } [0, 1], \quad \forall i \in \{1, 2, \ldots, N\}.
\end{aligned}
\tag{6}
$$

Where $x_i$ is the decision variable associated with set $i$, $c_i$ is the cost associated with selecting set $i$, and $S_i$ is the subset of items covered by set $i$. a new constraint ensures that each item $j$ is covered by at least one selected set, while the second constraint bounds the decision variables between 0 and 1. Some decision variables are binary, meaning sets must be fully selected or excluded, while others can take continuous values, allowing partial selection. The objective is to minimize the total cost of the selected sets while ensuring that every item is covered by at least one set. This mixed-integer formulation is particularly useful in scenarios where full or partial inclusion of resources is possible, providing a more flexible and realistic solution to resource allocation problems.

**Balanced Item Placement problem**: The Balanced Item Placement (BIP) problem arises in scenarios where items must be distributed across a set of buckets in such a way that the total load is balanced across the buckets. Each bucket has multiple resource constraints, and the goal is to minimize the imbalance in the bucket loads across these resource dimensions. This problem is commonly encountered in logistics, load balancing in distributed systems, and resource allocation in

cloud computing, where items (or tasks) need to be placed into buckets (or servers) under capacity constraints. Given $N$ items and $B$ buckets, each bucket has $R$ resource dimensions (such as weight, volume, or processing time). The objective is to distribute the items such that the maximum imbalance (deficit) across resource dimensions is minimized. The deficit in a particular dimension refers to the difference between the total resource used in that dimension and the target balanced load for that dimension across all buckets. Let $x_{i,j}$ be a binary decision variable where $x_{i,j} = 1$ if item $i$ is placed in bucket $j$, and $x_{i,j} = 0$ otherwise. Let $\text{deficit}_{j,r}$ represent the deficit for bucket $j$ in resource dimension $r$, and let $\text{max\_deficit}_r$ represent the maximum deficit across all buckets for resource dimension $r$. The BIP problem is formulated as follows:

$$
\begin{aligned}
\min \quad & \sum_{r=1}^{R} \text{max\_deficit}_r \\
\text{s.t.} \quad & \sum_{j=1}^{B} x_{i,j} = 1, \quad \forall i \in \{1, 2, \ldots, N\}, \\
& \sum_{i=1}^{N} w_{i,j,r} x_{i,j} \leq C_{j,r}, \quad \forall j \in \{1, 2, \ldots, B\}, \forall r \in \{1, 2, \ldots, R\}, \\
& \text{deficit}_{j,r} = \frac{\sum_{i=1}^{N} w_{i,j,r} x_{i,j}}{W_r}, \quad \forall j \in \{1, 2, \ldots, B\}, \forall r \in \{1, 2, \ldots, R\}, \\
& \text{max\_deficit}_r \geq \text{deficit}_{j,r}, \quad \forall j \in \{1, 2, \ldots, B\}, \forall r \in \{1, 2, \ldots, R\}, \\
& x_{i,j} \in \{0, 1\}, \quad \forall i \in \{1, 2, \ldots, N\}, \forall j \in \{1, 2, \ldots, B\}, \\
& \text{deficit}_{j,r}, \text{max\_deficit}_r \geq 0, \quad \forall j \in \{1, 2, \ldots, B\}, \forall r \in \{1, 2, \ldots, R\}.
\end{aligned}
\tag{7}
$$

Where $x_{i,j}$ is a binary decision variable indicating whether item $i$ is placed in bucket $j$; $w_{i,j,r}$ is the weight (or resource consumption) of item $i$ in bucket $j$ for resource dimension $r$; $C_{j,r}$ is the capacity of bucket $j$ in resource dimension $r$; $W_r$ is the total weight across all buckets for dimension $r$ (used for normalization); $\text{deficit}_{j,r}$ measures the imbalance in resource usage for bucket $j$ in dimension $r$; and $\text{max\_deficit}_r$ is the maximum deficit across all buckets for dimension $r$. The objective is to minimize the sum of the maximum deficits across all resource dimensions, ensuring that the load is as balanced as possible across buckets for each resource dimension.

**Combinatorial Auction Problem**: The Combinatorial Auction (CA) problem arises in auctions where bidders can place bids on combinations of items, rather than on individual items. The auctioneer's goal is to select a combination of bids that maximizes the total revenue while ensuring that each item is allocated to at most one bidder. This problem is commonly encountered in spectrum auctions, logistics, and online marketplaces, where items are complementary, and bidders value combinations of items more than individual ones. Given $N$ bids and $M$ items, each bid corresponds to a subset of items and has an associated value. The objective is to select a set of bids that maximizes the total value, subject to the constraint that each item can be allocated to at most one bid. Let $x_i$ represent a binary decision variable where $x_i = 1$ if bid $i$ is selected and $x_i = 0$ otherwise. The problem can be formulated as follows:

$$
\begin{aligned}
\max \quad & \sum_{i=1}^{N} c_i x_i \\
\text{s.t.} \quad & \sum_{i:j \in S_i} x_i \leq 1, \quad \forall j \in \{1, 2, \ldots, M\}, \\
& x_i \in \{0, 1\}, \quad \forall i \in \{1, 2, \ldots, N\}.
\end{aligned}
\tag{8}
$$

Where $x_i$ is a binary decision variable indicating whether bid $i$ is selected, $c_i$ is the value (or cost) associated with bid $i$, and $S_i$ is the subset of items associated with bid $i$. a new constraint ensures that each item $j$ is allocated to at most one bid by restricting the sum of the selected bids that include item $j$ to be less than or equal to 1. The objective is to maximize the total value of the selected bids while ensuring that no item is allocated to more than one bidder. In this problem, each bid can be

interpreted as a subset of items, and the goal is to find the optimal set of non-overlapping bids that yields the maximum revenue.

**Capacitated Facility Location Problem**: The Capacitated Facility Location (CFL) problem is a classical optimization problem in logistics and operations research. The objective is to determine the optimal locations for facilities (such as factories or warehouses) and the allocation of customers to these facilities, while respecting the capacity constraints of each facility and minimizing the total cost. In this problem, there are $N$ customers, $M$ facilities, and $K$ possible connections between customers and facilities. Each facility has a limited capacity, and each customer has a demand that must be met by one or more facilities. The aim is to minimize the overall cost, which includes the cost of opening facilities and the cost of assigning customers to open facilities. Let $x_{i,j}$ be a binary decision variable where $x_{i,j} = 1$ if customer $i$ is assigned to facility $j$, and $y_j = 1$ if facility $j$ is opened. The problem can be formulated as follows:

$$
\begin{aligned}
\min \quad & \sum_{i=1}^{N}\sum_{j=1}^{M} c_{i,j}x_{i,j} + \sum_{j=1}^{M} f_j y_j \\
\text{s.t.} \quad & \sum_{j=1}^{M} x_{i,j} = 1, \quad \forall i \in \{1, 2, \ldots, N\}, \\
& \sum_{i=1}^{N} d_i x_{i,j} \leq C_j y_j, \quad \forall j \in \{1, 2, \ldots, M\}, \\
& x_{i,j} \in \{0, 1\}, \quad \forall i \in \{1, 2, \ldots, N\}, \forall j \in \{1, 2, \ldots, M\}, \\
& y_j \in \{0, 1\}, \quad \forall j \in \{1, 2, \ldots, M\}.
\end{aligned}
\tag{9}
$$

Where $x_{i,j}$ is a binary decision variable denoting whether customer $i$ is assigned to facility $j$, and $y_j$ is a binary decision variable indicating whether facility $j$ is opened. The objective function minimizes the total cost, which includes the assignment costs $c_{i,j}$ of connecting customer $i$ to facility $j$, and the fixed costs $f_j$ of opening facility $j$. a new set of constraints ensures that each customer $i$ is assigned to exactly one facility. The second set of constraints ensures that the total demand of customers assigned to facility $j$ does not exceed its capacity $C_j$ if the facility is opened (i.e., if $y_j = 1$). The third and fourth constraints enforce the binary nature of the decision variables, meaning that customer assignments and facility openings are either fully made or not made.

**Middle-Mile Consolidation with Waiting Times**: The Middle-Mile Consolidation problem shows as follows.

$$
\begin{aligned}
\min \quad & \sum_{r \in \mathcal{R}} C_r x_r + \sum_{l \in \mathcal{L}}\sum_{m \in \mathcal{M}_l} (A_{lm}f_{lm} + B_{lm}v_{lm}) \\
\text{s.t.} \quad & \sum_{r \in \mathcal{R}_k} x_r = 1, \quad \forall k \in \mathcal{K}, \\
& \sum_{m \in \mathcal{M}_l} v_{lm} = \sum_{k \in \mathcal{K}} V_k x_r, \quad \forall l \in \mathcal{L}, \\
& v_{lm} \leq Q_{lm}^{\max} f_{lm}, \quad \forall l \in \mathcal{L}, \forall m \in \mathcal{M}_l, \\
& v_{lm} \geq Q_{lm}^{\min} f_{lm}, \quad \forall l \in \mathcal{L}, \forall m \in \mathcal{M}_l, \\
& \sum_{m \in \mathcal{M}_l} y_{lm} \leq 1, \quad \forall l \in \mathcal{L}, \\
& f_{lm} \leq F_{lm} y_{lm}, \quad \forall l \in \mathcal{L}, \forall m \in \mathcal{M}_l, \\
& x_r \in \{0, 1\}, \quad \forall r \in \mathcal{R}, \\
& y_{lm} \in \{0, 1\}, \quad \forall l \in \mathcal{L}, \forall m \in \mathcal{M}_l, \\
& f_{lm} \in \mathbb{Z}_{\geq 0}, \quad \forall l \in \mathcal{L}, \forall m \in \mathcal{M}_l, \\
& v_{lm} \geq 0, \quad \forall l \in \mathcal{L}, \forall m \in \mathcal{M}_l.
\end{aligned}
\tag{10}
$$

It focuses on selecting appropriate freight routes and determining load dispatch frequencies to minimize total transportation costs while ensuring that all commodity volumes are transported feasibly.

Shipment lead times are determined by the legs and transfer terminals along each route. The goal is to select a joint set of freight routes and load dispatch frequencies such that all commodity volumes are transported while minimizing the overall transportation cost. In this formulation, the cost to be minimized includes the handling cost for selecting routes and the cost of load dispatches and volumes transported on each lane. The constraints ensure that each commodity is assigned exactly one route, volumes on each lane are correctly allocated, and dispatches fall within the required capacity limits.

**Steiner Network Problem with Coverage Constraints**: The Steiner Network Problem with Coverage Constraints is a variation of the classical Steiner Tree Problem, where we seek to connect a set of terminal nodes in a graph such that certain coverage and flow constraints are met. The problem can be formulated as the following flow-based Mixed-Integer Linear Program (MILP). The goal is to minimize the total cost of selected edges while ensuring that coverage constraints are satisfied.

$$
\begin{aligned}
\min \quad & \sum_{(i,j)\in E} c(i,j)(x_{i,j} + x_{j,i}) \\
\text{s.t.} \quad & \sum_{e\in\delta^-(v)} y_e = 1[v \in C] + \sum_{e\in\delta^+(v)} (y_e + x_e), \quad \forall v \in V, \\
& x_{i,j} + x_{j,i} \leq 1, \quad \forall (i,j) \in E, \\
& \sum_{(i,j)\in S(r)} (x_{i,j} + x_{j,i}) \geq 1, \quad \forall r \in R, \\
& 0 \leq y_e \leq (|\hat{E}| + |V|)x_e, \quad \forall e \in \hat{E}, \\
& z + \sum_{t\in T} y_{0,t} = |\hat{E}| + |V|, \\
& \sum_{t\in T} y_{0,t} = |C| + \sum_{e\in\hat{E}} x_e, \\
& x_e \in \{0,1\}, \quad \forall e \in \hat{E}.
\end{aligned}
\tag{11}
$$

In this formulation, the objective function minimizes the total cost of selected edges. a new constraint ensures that flow conservation is maintained at each node, with nodes absorbing flow if they are in the coverage set $C$. The second constraint ensures that each edge between nodes $i$ and $j$ is selected at most once. The third constraint guarantees that at least one edge from each set $S(r)$ is selected. The fourth constraint relates the flow on edge $e$ to its selection. The fifth and sixth constraints ensure that the total flow injected into the system corresponds to the required flow to cover the nodes in $C$ and selected edges. Finally, the decision variables $x_e$ are binary, indicating whether an edge is selected or not.

As shown in Table 6, for each of the nine problem types, we generated instances of three different scales (easy, medium, and hard) to ensure that our benchmarks cover a wide range of complexities and sizes. Each scale is designed to accommodate different computational capacities and research needs, including large-scale problems. The table below summarizes the number of instances, average number of variables, and average number of constraints for each problem type and scale.

### A.4 Training and Testing Dataset Partition

For training and testing, we have randomly partitioned the problem data into training and testing datasets, the result of testing dataset is shown in Table 7 and Table 8.

## B Details of Algorithm

### B.1 Adaptive Constraint Partition Based Optimization Framework

Starting with the initial feasible solution gained from Gurobi, the Adaptive Constraint Partition Based Optimization Framework (ACP) attempts to iteratively enhance the current solution. Initially, constraints are randomly partitioned into disjoint blocks. In each iteration, only one block is considered, and the variables within this block are optimized while the values of other variables are fixed at

| Name (Path) | Number of Instances | Avg. Vars | Avg. Constraints |
|---|---|---|---|
| MIS_easy | 50 | 20000 | 60000 |
| MIS_medium | 50 | 100000 | 300000 |
| MIS_hard | 50 | 1000000 | 3000000 |
| MVC_easy | 50 | 20000 | 60000 |
| MVC_medium | 50 | 100000 | 300000 |
| MVC_hard | 50 | 1000000 | 3000000 |
| SC_easy | 50 | 40000 | 40000 |
| SC_medium | 50 | 200000 | 200000 |
| SC_hard | 50 | 2000000 | 2000000 |
| BIP_easy | 50 | 4081 | 290 |
| BIP_medium | 50 | 14182 | 690 |
| BIP_hard | 50 | 54584 | 2090 |
| CAT_easy | 50 | 2000 | 2000 |
| CAT_medium | 50 | 22000 | 22000 |
| CAT_hard | 50 | 2000000 | 2000000 |
| CFL_easy | 50 | 16040 | 80 |
| CFL_medium | 50 | 144200 | 320 |
| CFL_hard | 50 | 656520 | 800 |
| MIKS_easy | 50 | 5000 | 5000 |
| MIKS_medium | 50 | 55000 | 55000 |
| MIKS_hard | 50 | 1000000 | 1000000 |
| MMCW_easy | 50 | 5760 | 2880 |
| MMCW_medium | 50 | 55260 | 27630 |
| MMCW_hard | 50 | 253980 | 126990 |
| SNPCC_easy | 50 | 3000 | 30 |
| SNPCC_medium | 50 | 15000 | 151 |
| SNPCC_hard | 50 | 240000 | 2405 |

Table 6: Summary of Generated Instances for Nine Problem Types

the levels of the current optimal feasible solution. The number of blocks is adaptively updated based on the objective value improvement observed over the last two iterations, using a preset threshold. After the designated time (e.g., 8 hours), ACP terminates, and the improved solution $x^*$ is used to update the Gurobi-derived solution $\overline{x}$ and is packaged into a pickle file along with the problem instance.

### B.2 SIMILARITY EVALUATION METRICS

Specifically, we detail an embedding approach representing MILP instances as a 10-dimensional embedding, capturing fundamental aspects of the mathematical formulation and bipartite graph characteristics. This vector includes metrics such as Fraction of non-zero entries in the coefficient matrix, Mean and standard deviation of the degrees of constraint vertices, Mean and standard deviation of the degrees of variable vertices, Mean and standard deviation of non-zero entries in the coefficient matrix, Mean and standard deviation of RHS values, and Modularity of the bipartite graph representation.

It is noted that, using embeddings alone is not sufficient to determine the isomorphism of datasets. To ensure the robustness of our classification, we employed two levels of similarity evaluation metrics. a new metric assesses the structural similarity of problem instances within the dataset, while the second metric evaluates the similarity at the embedding level. These two metrics complement each other, providing a more comprehensive assessment of the dataset's classification quality. Figure 5 in the main text illustrates the similarity of subproblems after classification, showing that structural and embedding similarities corroborate each other, indicating a degree of robustness. Additionally, the experimental results discussed in Figure 6 and 7, comparing before and after reclassification, further validate the effectiveness of our graph classification approach.

## C DETAILS OF EXPERIMENTS

### C.1 EXPERIMENTS ENVIRONMENTS

All experiments are run on a machine with Intel Xeon Platinum 8375C @ 2.90GHz CPU and four NVIDIA TESLA V100(32G) GPUs. Each scale of any Benchmark MILP is tested on several instances, and the results shown are the average of the five results.

| | | | |
|---|---|---|---|
| MIPLIB (Minimize) | 30_70_4.5_0.5_10.lp
30_70_4.5_0.5_100.lp
mkc1.lp
neos14.lp | anonymous (Minimize) | anonymous_5.lp
anonymous_29.lp
anonymous_39.lp
anonymous_76.lp |
| Coral (Minimize) | neos-548251.lp
neos-582605.lp
neos-584146.lp
neos-847302.lp
neos-935496.lp
neos-935674.lp
neos-1430811.lp | Transportation (Minimize) | n3701.lp
n3704.lp
n3705.lp
n3707.lp
n3709.lp |
| Cut (Minimize) | n3-3.lp
n9-3.lp
u15-3.lp | ECOGCNN (Minimize) | ns3337549.lp
ns4165869.lp
u40t24ramp.lp |
| item_placement (Minimize) | item_placement_0.lp
item_placement_3.lp
item_placement_18.lp
item_placement_24.lp
item_placement_27.lp
item_placement_34.lp
item_placement_35.lp
item_placement_45.lp
item_placement_46.lp
item_placement_49.lp | load_balancing (Minimize) | load_balancing_15.lp
load_balancing_16.lp
load_balancing_18.lp
load_balancing_30.lp
load_balancing_34.lp
load_balancing_61.lp
load_balancing_67.lp
load_balancing_73.lp
load_balancing_80.lp
load_balancing_87.lp |
| Nexp (Minimize) | germanrr.lp
p6b.lp
ramos3.lp
seymour.disj-10.lp
sp98ar.lp | vary_bounds_s1 (Minimize) | bnd_s1_i01.lp
bnd_s1_i02.lp
bnd_s1_i03.lp
bnd_s1_i04.lp
bnd_s1_i05.lp |
| vary_bounds_s2 (Minimize) | bnd_s2_i01.lp
bnd_s2_i02.lp
bnd_s2_i03.lp
bnd_s2_i04.lp
bnd_s2_i05.lp | vary_bounds_s3 (Minimize) | bnd_s3_i01.lp
bnd_s3_i02.lp
bnd_s3_i03.lp
bnd_s3_i04.lp
bnd_s3_i05.lp |
| vary_matrix_s1 (Minimize) | mat_s1_i01.lp
mat_s1_i02.lp
mat_s1_i03.lp
mat_s1_i04.lp
mat_s1_i05.lp | vary_matrix_rhs_bounds_s1 (Minimize) | mat_rhs_bnd_s1_i09.lp
mat_rhs_bnd_s1_i17.lp
mat_rhs_bnd_s1_i33.lp
mat_rhs_bnd_s1_i43.lp
mat_rhs_bnd_s1_i47.lp |
| vary_matrix_rhs_bounds_obj (Minimize) | mat_rhs_bnd_obj_s1_i01.lp
mat_rhs_bnd_obj_s1_i02.lp
mat_rhs_bnd_obj_s1_i04.lp
mat_rhs_bnd_obj_s1_i15.lp
mat_rhs_bnd_obj_s1_i38.lp | vary_obj_s1 (Minimize) | obj_s1_i01.lp
obj_s1_i02.lp
obj_s1_i03.lp
obj_s1_i04.lp
obj_s1_i05.lp |
| vary_obj_s2 (Minimize) | obj_s2_i11.lp
obj_s2_i21.lp
obj_s2_i22.lp
obj_s2_i44.lp
obj_s2_i49.lp | vary_obj_s3 (Minimize) | obj_s3_i01.lp
obj_s3_i02.lp
obj_s3_i03.lp
obj_s3_i04.lp
obj_s3_i05.lp |
| vary_rhs_s1 (Minimize) | rhs_s1_i01.lp
rhs_s1_i02.lp
rhs_s1_i03.lp
rhs_s1_i04.lp
rhs_s1_i36.lp | vary_rhs_s2 (Minimize) | rhs_s2_i01.lp
rhs_s2_i02.lp
rhs_s2_i03.lp
rhs_s2_i04.lp
rhs_s2_i05.lp |
| vary_rhs_s3 (Minimize) | rhs_s3_i01.lp
rhs_s3_i02.lp
rhs_s3_i33.lp
rhs_s3_i35.lp
rhs_s3_i39.lp | vary_rhs_s4 (Minimize) | rhs_s4_i01.lp
rhs_s4_i02.lp
rhs_s4_i03.lp
rhs_s4_i04.lp
rhs_s4_i05.lp |

Table 7: The result of testing dataset partition, Part A.

| vary_rhs_obj_s1 (Minimize) | rhs_obj_s1_i19.lp
rhs_obj_s1_i21.lp
rhs_obj_s1_i27.lp
rhs_obj_s1_i46.lp
rhs_obj_s1_i48.lp | vary_rhs_obj_s2 (Minimize) | rhs_obj_s2_i01.lp
rhs_obj_s2_i13.lp
rhs_obj_s2_i18.lp
rhs_obj_s2_i26.lp
rhs_obj_s2_i43.lp |
|---|---|---|---|
| IS_easy (Maximize) | IS_easy_instance_5.lp
IS_easy_instance_7.lp
IS_easy_instance_9.lp
IS_easy_instance_14.lp
IS_easy_instance_21.lp | IS_medium (Maximize) | IS_medium_instance_0.lp
IS_medium_instance_8.lp
IS_medium_instance_11.lp
IS_medium_instance_22.lp
IS_medium_instance_29.lp |
| IS_hard (Maximize) | IS_hard_instance_2.lp
IS_hard_instance_3.lp
IS_hard_instance_7.lp
IS_hard_instance_15.lp
IS_hard_instance_16.lp | fc.data (Minimize) | fc.30.50.1.lp
fc.30.50.2.lp
fc.30.50.3.lp
fc.30.50.4.lp
fc.30.50.10.lp |
| nn_verification (Maximize) | test_120.proto.lp
test_1082.proto.lp
test_1087.proto.lp
test_1112.proto.lp
test_1116.proto.lp
test_1231.proto.lp
test_1987.proto.lp
test_5470.proto.lp
training_47482.proto.lp | Aclib (Minimize) | cls.T90.C2.F500.S2.lp
cls.T90.C2.F500.S3.lp
cls.T90.C3.F100.S5.lp
cls.T90.C3.F500.S2.lp
cls.T90.C3.F500.S5.lp
cls.T90.C4.F100.S4.lp
cls.T90.C4.F500.S4.lp
cls.T90.C4.F1000.S1.lp
cls.T90.C5.F250.S4.lp
cls.T90.C5.F1000.S2.lp |
| MVC_easy (Minimize) | MVC_easy_instance_1.lp
MVC_easy_instance_14.lp
MVC_easy_instance_16.lp
MVC_easy_instance_19.lp
MVC_easy_instance_22.lp | MVC_medium (Minimize) | MVC_medium_instance_0.lp
MVC_medium_instance_4.lp
MVC_medium_instance_9.lp
MVC_medium_instance_11.lp
MVC_medium_instance_22.lp |
| MVC_hard (Minimize) | MVC_hard_instance_2.lp
MVC_hard_instance_5.lp
MVC_hard_instance_6.lp
MVC_hard_instance_27.lp
MVC_hard_instance_28.lp | SC_easy (Minimize) | SC_easy_instance_3.lp
SC_easy_instance_9.lp
SC_easy_instance_12.lp
SC_easy_instance_22.lp
SC_easy_instance_26.lp |
| SC_medium (Minimize) | SC_medium_instance_4.lp
SC_medium_instance_5.lp
SC_medium_instance_10.lp
SC_medium_instance_16.lp
SC_medium_instance_18.lp | SC_hard (Minimize) | SC_hard_instance_7.lp
SC_hard_instance_13.lp
SC_hard_instance_21.lp
SC_hard_instance_25.lp
SC_hard_instance_27.lp |
| HEM_knapsack (Maximize) | instance_46.lp
instance_152.lp
instance_270.lp
instance_864.lp
instance_875.lp | HEM_mis (Maximize) | instance_119.lp
instance_372.lp
instance_557.lp
instance_848.lp
instance_855.lp |
| HEM_setcover (Minimize) | instance_22.lp
instance_446.lp
instance_521.lp
instance_762.lp
instance_997.lp | HEM_corlat (Maximize) | cor-lat-2f+r-u-10-10-10-5-100-3.003.b78.000000.prune2.lp
cor-lat-2f+r-u-10-10-10-5-100-3.003.b85.000000.prune2.lp
cor-lat-2f+r-u-10-10-10-5-100-3.007.b78.000000.prune2.lp
cor-lat-2f+r-u-10-10-10-5-100-3.007.b660.000000.prune2.lp
cor-lat-2f+r-u-10-10-10-5-100-3.008.b92.000000.prune2.lp |
| HEM_mik (Minimize) | mik.250-10-50.1.lp
mik.250-10-100.1.lp
mik.250-10-100.3.lp
mik.250-10-100.5.lp
mik.250-20-100.3.lp | | |

Table 8: The result of testing dataset partition, Part B.

## C.2 DATASET ANALYSIS

It involved measuring both graph structural embedding distances and neural embedding distances and comparing these metrics with those from established MILP datasets. We restricted our comparison to representative problems containing fewer than 50,000 decision variables to manage computational demands.

## C.3 DATASET RECLASSIFICATION

For some datasets, especially those collected from open-source datasets, the problem instances generated from various scenarios are often mixed, leading to a highly complex distribution. Therefore, we employed the spectral clustering algorithm to categorize the complete datasets. The result is shown as follows.

**MIPlib**:

- Cluster 0: []
- Cluster 1: ['swath2.pkl', 'neos9.pkl', 'swath1.pkl', 'swath3.pkl', 'neos818918.pkl', 'neos648910.pkl', 'mkc1.pkl']
- Cluster 2: ['bienst1.pkl', 'bienst2.pkl']
- Cluster 3: ['ns183.pkl', '30_70_45_05_100.pkl', 'seymour1.pkl', '30_70_45_095_100.pkl', 'dano3_4.pkl', '30_70_45_05_10.pkl', 'neos17.pkl', 'dano3_3.pkl', 'ns1830653.pkl', '30_70_45_095_98.pkl', 'dano3_5.pkl']
- Cluster 4: ['neos3.pkl', 'neos12.pkl', 'neos23.pkl', 'neos808444.pkl', 'bc1.pkl', 'neos2.pkl', 'neos1.pkl', 'neos22.pkl', 'neos11.pkl', 'neos6.pkl', 'markshare_5_0.pkl', 'neos823206.pkl']
- Cluster 5: ['neos8.pkl', 'neos10.pkl']
- Cluster 6: []
- Cluster 7: ['neos5.pkl', 'ran14x18_1.pkl', 'ns1648184.pkl', 'neos20.pkl', 'neos21.pkl', 'ns1688347.pkl', 'nug08.pkl', 'ns1692855.pkl', 'neos897005.pkl', 'neos7.pkl', 'neos13.pkl', 'markshare_4_0.pkl', 'neos14.pkl', 'qap10.pkl', 'neos4.pkl', 'ns1671066.pkl']

**Coral**:

- Cluster 0: ['neos-841664.pkl', 'neos-796608.pkl', 'leo1.pkl', 'neos-1112782.pkl', 'neos-1053234.pkl', 'neos-631517.pkl', 'neos-1171448.pkl', 'neos-584146.pkl', 'neos-885524.pkl', 'neos-1171692.pkl', 'neos-801834.pkl', 'neos-848198.pkl', 'neos-1211578.pkl', 'neos-1324574.pkl', 'neos-1228986.pkl', 'neos-504674.pkl', 'neos-619167.pkl', 'neos-933550.pkl', 'neos-1096528.pkl', 'neos-807639.pkl', 'neos-810286.pkl', 'neos-693347.pkl', 'neos-934531.pkl', 'neos-960392.pkl', 'neos-906865.pkl', 'd20200.pkl', 'neos-1367061.pkl', 'binkar10_1.pkl', 'neos-1224597.pkl', 'neos-582605.pkl', 'neos-955800.pkl', 'neos-1056905.pkl', 'neos-595925.pkl', 'neos-548251.pkl', 'neos-1061020.pkl', 'neos-885086.pkl', 'neos-1425699.pkl']
- Cluster 1: ['neos-785899.pkl', 'neos-570431.pkl', 'neos-933364.pkl', 'neos-933815.pkl']
- Cluster 2: ['neos-785914.pkl', 'neos-1413153.pkl', 'neos-555001.pkl', 'neos-551991.pkl', 'neos-555884.pkl', 'neos-1415183.pkl', 'neos-631784.pkl', 'neos-738098.pkl', 'neos-578379.pkl']
- Cluster 3: ['neos-941717.pkl', 'neos-1440225.pkl', 'neos-1208069.pkl', 'neos-1151496.pkl', 'neos-848845.pkl', 'neos-912023.pkl', 'neos-847302.pkl', 'neos-808214.pkl']
- Cluster 4: ['neos-948268.pkl', 'neos-829552.pkl', 'neos-501453.pkl', 'neos-1427181.pkl', 'bienst1.pkl', 'neos-611838.pkl', 'neos-1427261.pkl', 'mcsched.pkl', 'neos-1436713.pkl', 'neos-612162.pkl', 'bienst2.pkl', 'neos-1429185.pkl']
- Cluster 5: []
- Cluster 6: []
- Cluster 7: ['aligninq.pkl', 'neos-595905.pkl', 'neos-791021.pkl', 'neos-691073.pkl', 'neos-957323.pkl', 'neos-916173.pkl', 'neos-1330635.pkl', 'neos-494568.pkl', 'neos-555771.pkl', 'neos-935348.pkl', 'neos-936660.pkl', 'neos-954925.pkl', 'neos-824695.pkl', 'neos-1420546.pkl', 'neos-937815.pkl', 'neos-1120495.pkl', 'neos-1311124.pkl', 'dano3_3.pkl', 'neos-826812.pkl', 'neos-935769.pkl', 'neos-1429461.pkl', 'neos-585192.pkl', 'neos-530627.pkl', 'neos-662469.pkl', 'neos-476283.pkl', 'neos-498623.pkl', 'neos-1058477.pkl', 'neos-480878.pkl', 'neos-633273.pkl', 'neos-799711.pkl', 'neos-1426662.pkl', 'neos-803219.pkl', 'neos-1440447.pkl', 'neos-941262.pkl', 'neos-565815.pkl', 'neos-826841.pkl', 'neos-522351.pkl', 'neos-937511.pkl']

**Nexp**:

- Cluster 0: ['neos-1603512.pkl', 'fiball.pkl', 'bienst1.pkl', 'newdano.pkl', 'neos16.pkl', 'neos-1603518.pkl', 'neos14.pkl', 'neos15.pkl', 'bienst2.pkl', 'mkc1.pkl']

- Cluster 1: ['neos-1622252.pkl', 'neos-1599274.pkl', 'ran14x18disj-8.pkl', 'neos20.pkl', 'neos-1603965.pkl', 'rlp1.pkl', 'p6b.pkl', 'neos-1616732.pkl', 'seymourdisj-10.pkl', 'pg5_34.pkl', 'd20200.pkl', 'neos4.pkl', 'neos-1620770.pkl', 'mcf2.pkl', 'prod2.pkl']

- Cluster 2: ['leo1.pkl', 'neos12.pkl', 'nug08.pkl', 'neos1.pkl', 'neos-1620807.pkl', 'neos11.pkl', 'neos7.pkl', 'neos-1593097.pkl', 'roy.pkl', 'neos-1595230.pkl', 'qap10.pkl', 'neos18.pkl', 'neos-1582420.pkl']

- Cluster 3: ['neos5.pkl', 'neos-1605061.pkl', 'ramos3.pkl', 'aligninq.pkl', 'sp97ic.pkl', 'neos-1605075.pkl', 'leo2.pkl', 'neos-1601936.pkl', 'mcsched.pkl', 'sp98ir.pkl', 'sp98ar.pkl']

- Cluster 4: ['prod1.pkl', 'haprp.pkl', 'neos17.pkl', 'lrn.pkl', 'nsa.pkl', 'neos-1516309.pkl', 'pg.pkl']

- Cluster 5: ['22433.pkl', 'd1020.pkl', 'dano3_4.pkl', '23588.pkl', 'dano3_3.pkl', 'dano3_5.pkl', 'd10200.pkl']

- Cluster 6: ['neos3.pkl', 'ran14x18_1.pkl', 'neos2.pkl']

- Cluster 7: []

**ECOGCNN**:

- Cluster 0: ['ns2382816.pkl', 'ns3633010.pkl', 'ns2081729.pkl', 'ns2394796.pkl', 'neos-849702.pkl', 'ns3134812.pkl']

- Cluster 1: ['ns43503.pkl', 'ns2164569.pkl']

- Cluster 2: ['ns2082847.pkl', 'u50t24wc.pkl', 'ns2369235.pkl', 'ns2996139.pkl', 'ns2070961.pkl', 'ns4636843.pkl']

- Cluster 3: ['ns2326618.pkl', 'ns2494475.pkl', 'ns3337549.pkl', 'ns1943024.pkl', 'ns2071214.pkl', 'u30t24ramp.pkl']

- Cluster 4: ['ns2267839.pkl', 'ns2350781.pkl']

- Cluster 5: ['chrom_512.pkl', 'chrom_256.pkl', 'u40t24ramp.pkl', 'u40t24wc.pkl']

- Cluster 6: []

- Cluster 7: []

## C.4 SETTINGS OF BENCHMARKING STUDY

In our benchmarking study, hyperparameter selection played a crucial role in optimizing the performance of the baseline algorithms. We employed different strategies for hyperparameter tuning across the various methods tested in this study. While some algorithms performed well with default settings, others required manual tuning of specific hyperparameters to improve their performance on complex problem instances.

We evaluated six baseline algorithms in the main text: Gurobi, SCIP, LNS, ACP, Learn2Branch, and GNN&GBDT. For Gurobi and SCIP, we used the default solver settings, as these configurations are generally well-optimized for a wide range of problems and provide strong performance without requiring further adjustments.

For the other algorithms—LNS, ACP, Learn2Branch, and GNN&GBDT—manual hyperparameter tuning was necessary. This tuning process involved adjusting key hyperparameters, such as learning rates, batch sizes, and neural network architectures, depending on the algorithm and the specific problem type. Our goal was to achieve better performance, particularly on more complex and challenging problem instances.

In addition to the six baseline algorithms, we also included four additional machine learning-based algorithms: Neural Diving, Predict&Search, Hybrid_Learn2Branch, and GNN-MILP. Similar to the previously mentioned methods, we applied careful hyperparameter tuning for these algorithms to ensure optimal performance. The default settings of Gurobi and SCIP were retained, while the other eight algorithms underwent a systematic tuning process to improve their results.

The detailed hyperparameter settings for each algorithm are provided in Tables 9 through 17, which summarize the configurations used for each method. These tables offer a comprehensive overview of

the hyperparameters and their respective values, along with the rationale behind any modifications. This information is essential for understanding the choices made during the tuning process and for facilitating the reproducibility of our results across different problem instances.

| Problem | blocking num |
|---|---|
| MIS_easy | 2 |
| MIS_medium | 4 |
| MIS_hard | 6 |
| MCV_easy | 3 |
| MVC_medium | 4 |
| MVC_hard | 5 |
| SC_easy | 2 |
| SC_medium | 4 |
| SC_hard | 6 |
| MIPlib | 2 |
| Coral | 2 |
| Cut | 2 |
| ECOGCNN | 2 |
| HEM_knapsack | 2 |
| HEM_mis | 2 |
| HEM_setcover | 2 |
| HEM_corlat | 2 |
| HEM_mik | 2 |
| item_placement | 3 |
| load_balancing | 3 |
| anonymous | 3 |
| Nexp | 2 |
| Transportation | 2 |
| vary_bounds_s1 | 2 |
| vary_bounds_s2 | 2 |
| vary_bounds_s3 | 2 |
| vary_matrix_s1 | 2 |
| vary_matrix_rhs_bounds_s1 | 2 |
| vary_matrix_rhs_bounds_obj_s1 | 2 |
| vary_obj_s1 | 2 |
| vary_obj_s2 | 2 |
| vary_obj_s3 | 2 |
| vary_rhs_s1 | 2 |
| vary_rhs_s2 | 2 |
| vary_rhs_s3 | 2 |
| vary_rhs_s4 | 2 |
| vary_rhs_obj_s1 | 2 |
| vary_rhs_obj_s2 | 2 |
| Aclib | 2 |
| fc.data | 2 |
| nn_verification | 2 |

| Problem | initial blocking num |
|---|---|
| MIS_easy | 2 |
| MIS_medium | 2 |
| MIS_hard | 2 |
| MCV_easy | 2 |
| MVC_medium | 2 |
| MVC_hard | 2 |
| SC_easy | 2 |
| SC_medium | 2 |
| SC_hard | 2 |
| MIPlib | 2 |
| Coral | 2 |
| Cut | 2 |
| ECOGCNN | 2 |
| HEM_knapsack | 2 |
| HEM_mis | 2 |
| HEM_setcover | 2 |
| HEM_corlat | 2 |
| HEM_mik | 2 |
| item_placement | 2 |
| load_balancing | 2 |
| anonymous | 2 |
| Nexp | 2 |
| Transportation | 2 |
| vary_bounds_s1 | 2 |
| vary_bounds_s2 | 2 |
| vary_bounds_s3 | 2 |
| vary_matrix_s1 | 2 |
| vary_matrix_rhs_bounds_s1 | 2 |
| vary_matrix_rhs_bounds_obj_s1 | 2 |
| vary_obj_s1 | 2 |
| vary_obj_s2 | 2 |
| vary_obj_s3 | 2 |
| vary_rhs_s1 | 2 |
| vary_rhs_s2 | 2 |
| vary_rhs_s3 | 2 |
| vary_rhs_s4 | 2 |
| vary_rhs_obj_s1 | 2 |
| vary_rhs_obj_s2 | 2 |
| Aclib | 2 |
| fc.data | 2 |
| nn_verification | 2 |

Table 9: Hyperparameter selection for LNS(left) and ACP (right).

| Problem | sample-rate | learning-rate | max-epoch |
|---|---|---|---|
| MIPlib | 10 | 0.001 | 1000 |
| Coral | 10 | 0.001 | 1000 |
| Cut | 10 | 0.001 | 1000 |
| ECOGCNN | 10 | 0.001 | 1000 |
| HEM_knapsack | 10 | 0.001 | 1000 |
| HEM_mis | 10 | 0.001 | 1000 |
| HEM_setcover | 10 | 0.001 | 1000 |
| HEM_corlat | 10 | 0.001 | 1000 |
| HEM_mik | 10 | 0.001 | 1000 |
| item_placement | 10 | 0.001 | 1000 |
| load_balancing | 10 | 0.001 | 1000 |
| anonymous | 10 | 0.001 | 1000 |
| Nexp | 10 | 0.001 | 1000 |
| Transportation | 10 | 0.001 | 1000 |
| vary_bounds_s1 | 10 | 0.001 | 1000 |
| vary_bounds_s2 | 1 | 0.001 | 1000 |
| vary_bounds_s3 | 1 | 0.001 | 1000 |
| vary_matrix_s1 | 10 | 0.001 | 1000 |
| vary_matrix_rhs_bounds_s1 | 10 | 0.001 | 1000 |
| vary_matrix_rhs_bounds_obj_s1 | 10 | 0.001 | 1000 |
| vary_obj_s1 | 10 | 0.001 | 1000 |
| vary_obj_s2 | 10 | 0.001 | 1000 |
| vary_obj_s3 | 10 | 0.001 | 1000 |
| vary_rhs_s1 | 10 | 0.001 | 1000 |
| vary_rhs_s2 | 10 | 0.001 | 1000 |
| vary_rhs_s3 | 10 | 0.001 | 1000 |
| vary_rhs_s4 | 10 | 0.001 | 1000 |
| vary_rhs_obj_s1 | 10 | 0.001 | 1000 |
| vary_rhs_obj_s2 | 10 | 0.001 | 1000 |
| Aclib | 10 | 0.001 | 1000 |
| fc.data | 10 | 0.001 | 1000 |
| nn_verification | 10 | 0.001 | 1000 |

Table 10: Hyperparameter selection for Learn2Branch (Part 1).

| Problem | num-bad-epoch | batch-size | pretrain-batch-size | valid-batch-size |
|---|---|---|---|---|
| MIPlib | 20 | 64 | 64 | 64 |
| Coral | 20 | 64 | 64 | 64 |
| Cut | 20 | 64 | 64 | 64 |
| ECOGCNN | 20 | 64 | 64 | 64 |
| HEM_knapsack | 20 | 64 | 64 | 64 |
| HEM_mis | 20 | 64 | 64 | 64 |
| HEM_setcover | 20 | 64 | 64 | 64 |
| HEM_corlat | 20 | 64 | 64 | 64 |
| HEM_mik | 20 | 64 | 64 | 64 |
| item_placement | 20 | 64 | 64 | 64 |
| load_balancing | 20 | 64 | 64 | 64 |
| anonymous | 20 | 64 | 64 | 64 |
| Nexp | 20 | 64 | 64 | 64 |
| Transportation | 20 | 64 | 64 | 64 |
| vary_bounds_s1 | 20 | 64 | 64 | 64 |
| vary_bounds_s2 | 20 | 64 | 64 | 64 |
| vary_bounds_s3 | 20 | 64 | 64 | 64 |
| vary_matrix_s1 | 20 | 64 | 64 | 64 |
| vary_matrix_rhs_bounds_s1 | 20 | 64 | 64 | 64 |
| vary_matrix_rhs_bounds_obj_s1 | 20 | 64 | 64 | 64 |
| vary_obj_s1 | 20 | 64 | 64 | 64 |
| vary_obj_s2 | 20 | 64 | 64 | 64 |
| vary_obj_s3 | 20 | 64 | 64 | 64 |
| vary_rhs_s1 | 20 | 64 | 64 | 64 |
| vary_rhs_s2 | 20 | 64 | 64 | 64 |
| vary_rhs_s3 | 20 | 64 | 64 | 64 |
| vary_rhs_s4 | 20 | 64 | 64 | 64 |
| vary_rhs_obj_s1 | 20 | 64 | 64 | 64 |
| vary_rhs_obj_s2 | 20 | 64 | 64 | 64 |
| Aclib | 20 | 64 | 64 | 64 |
| fc.data | 20 | 64 | 64 | 64 |
| nn_verification | 20 | 64 | 64 | 64 |

Table 11: Hyperparameter selection for Learn2Branch (Part 2).

| Problem | learning-rate | max-patient-epoch | batch-size | n_estimators | max_depth | rate | fix |
|---|---|---|---|---|---|---|---|
| MIS_easy | 0.0001 | 10 | 1 | 30 | 5 | 0.4 | 0.6 |
| MIS_medium | 0.0001 | 10 | 1 | 30 | 5 | 0.4 | 0.6 |
| MIS_hard | 0.0001 | 10 | 1 | 30 | 5 | 0.4 | 0.6 |
| MCV_easy | 0.0001 | 10 | 1 | 30 | 5 | 0.4 | 0.6 |
| MVC_medium | 0.0001 | 10 | 1 | 30 | 5 | 0.4 | 0.6 |
| MVC_hard | 0.0001 | 10 | 1 | 30 | 5 | 0.4 | 0.6 |
| SC_easy | 0.0001 | 10 | 1 | 30 | 5 | 0.4 | 0.6 |
| SC_medium | 0.0001 | 10 | 1 | 30 | 5 | 0.4 | 0.6 |
| SC_hard | 0.0001 | 10 | 1 | 30 | 5 | 0.4 | 0.6 |
| MIPlib | 0.0001 | 10 | 1 | 30 | 5 | 0.4 | 0.6 |
| Coral | 0.0001 | 10 | 1 | 30 | 5 | 0.4 | 0.6 |
| Cut | 0.0001 | 10 | 1 | 30 | 5 | 0.4 | 0.6 |
| ECOGCNN | 0.0001 | 10 | 1 | 30 | 5 | 0.4 | 0.6 |
| HEM_knapsack | 0.0001 | 10 | 1 | 30 | 5 | 0.4 | 0.6 |
| HEM_mis | 0.0001 | 10 | 1 | 30 | 5 | 0.4 | 0.6 |
| HEM_setcover | 0.0001 | 10 | 1 | 30 | 5 | 0.4 | 0.6 |
| HEM_corlat | 0.0001 | 10 | 1 | 30 | 5 | 0.4 | 0.6 |
| HEM_mik | 0.0001 | 10 | 1 | 30 | 5 | 0.4 | 0.6 |
| item_placement | 0.0001 | 10 | 1 | 30 | 5 | 0.4 | 0.6 |
| load_balancing | 0.0001 | 10 | 1 | 30 | 5 | 0.4 | 0.6 |
| anonymous | 0.0001 | 10 | 1 | 30 | 5 | 0.4 | 0.6 |
| Nexp | 0.0001 | 10 | 1 | 30 | 5 | 0.4 | 0.6 |
| Transportation | 0.0001 | 10 | 1 | 30 | 5 | 0.4 | 0.6 |
| vary_bounds_s1 | 0.0001 | 10 | 1 | 30 | 5 | 0.4 | 0.6 |
| vary_bounds_s2 | 0.0001 | 10 | 1 | 30 | 5 | 0.4 | 0.6 |
| vary_bounds_s3 | 0.0001 | 10 | 1 | 30 | 5 | 0.4 | 0.6 |
| vary_matrix_s1 | 0.0001 | 10 | 1 | 30 | 5 | 0.4 | 0.6 |
| vary_matrix_rhs_bounds_s1 | 0.0001 | 10 | 1 | 30 | 5 | 0.4 | 0.6 |
| vary_matrix_rhs_bounds_obj_s1 | 0.0001 | 10 | 1 | 30 | 5 | 0.4 | 0.6 |
| vary_obj_s1 | 0.0001 | 10 | 1 | 30 | 5 | 0.4 | 0.6 |
| vary_obj_s2 | 0.0001 | 10 | 1 | 30 | 5 | 0.4 | 0.6 |
| vary_obj_s3 | 0.0001 | 10 | 1 | 30 | 5 | 0.4 | 0.6 |
| vary_rhs_s1 | 0.0001 | 10 | 1 | 30 | 5 | 0.4 | 0.6 |
| vary_rhs_s2 | 0.0001 | 10 | 1 | 30 | 5 | 0.4 | 0.6 |
| vary_rhs_s3 | 0.0001 | 10 | 1 | 30 | 5 | 0.4 | 0.6 |
| vary_rhs_s4 | 0.0001 | 10 | 1 | 30 | 5 | 0.4 | 0.6 |
| vary_rhs_obj_s1 | 0.0001 | 10 | 1 | 30 | 5 | 0.4 | 0.6 |
| vary_rhs_obj_s2 | 0.0001 | 10 | 1 | 30 | 5 | 0.4 | 0.6 |
| Aclib | 0.0001 | 10 | 1 | 30 | 5 | 0.4 | 0.6 |
| fc.data | 0.0001 | 10 | 1 | 30 | 5 | 0.4 | 0.6 |
| nn_verification | 0.0001 | 10 | 1 | 30 | 5 | 0.4 | 0.6 |

Table 12: Hyperparameter selection for GNN&GBDT.

## C.5 BENCHMARKING STUDY

To validate the effectiveness of ML4MILP and assess the performance of various baseline algorithms across different scenarios, we conducted a comprehensive benchmarking study. The evalu-

| Problem | learning-rate | NB-epochs | batch-size | weight-norm |
|---|---|---|---|---|
| MIS_easy | 0.001 | 9999 | 4 | 100 |
| MIS_medium | 0.001 | 9999 | 4 | 100 |
| MIS_hard | 0.001 | 9999 | 4 | 100 |
| MCV_easy | 0.001 | 9999 | 4 | 100 |
| MVC_medium | 0.001 | 9999 | 4 | 100 |
| MVC_hard | 0.001 | 9999 | 4 | 100 |
| SC_easy | 0.001 | 9999 | 4 | 100 |
| SC_medium | 0.001 | 9999 | 4 | 100 |
| SC_hard | 0.001 | 9999 | 4 | 100 |
| MIPlib | 0.001 | 9999 | 4 | 100 |
| Coral | 0.001 | 9999 | 4 | 100 |
| Cut | 0.001 | 9999 | 4 | 100 |
| ECOGCNN | 0.001 | 9999 | 4 | 100 |
| HEM_knapsack | 0.001 | 9999 | 4 | 100 |
| HEM_mis | 0.001 | 9999 | 4 | 100 |
| HEM_setcover | 0.001 | 9999 | 4 | 100 |
| HEM_corlat | 0.001 | 9999 | 4 | 100 |
| HEM_mik | 0.001 | 9999 | 4 | 100 |
| item_placement | 0.001 | 9999 | 4 | 100 |
| load_balancing | 0.001 | 9999 | 4 | 100 |
| anonymous | 0.001 | 9999 | 4 | 100 |
| Nexp | 0.001 | 9999 | 4 | 100 |
| Transportation | 0.001 | 9999 | 4 | 100 |
| vary_bounds_s1 | 0.001 | 9999 | 4 | 100 |
| vary_bounds_s2 | 0.001 | 9999 | 4 | 100 |
| vary_bounds_s3 | 0.001 | 9999 | 4 | 100 |
| vary_matrix_s1 | 0.001 | 9999 | 4 | 100 |
| vary_matrix_rhs_bounds_s1 | 0.001 | 9999 | 4 | 100 |
| vary_matrix_rhs_bounds_obj_s1 | 0.001 | 9999 | 4 | 100 |
| vary_obj_s1 | 0.001 | 9999 | 4 | 100 |
| vary_obj_s2 | 0.001 | 9999 | 4 | 100 |
| vary_obj_s3 | 0.001 | 9999 | 4 | 100 |
| vary_rhs_s1 | 0.001 | 9999 | 4 | 100 |
| vary_rhs_s2 | 0.001 | 9999 | 4 | 100 |
| vary_rhs_s3 | 0.001 | 9999 | 4 | 100 |
| vary_rhs_s4 | 0.001 | 9999 | 4 | 100 |
| vary_rhs_obj_s1 | 0.001 | 9999 | 4 | 100 |
| vary_rhs_obj_s2 | 0.001 | 9999 | 4 | 100 |
| Aclib | 0.001 | 9999 | 4 | 100 |
| fc.data | 0.001 | 9999 | 4 | 100 |
| nn_verification | 0.001 | 9999 | 4 | 100 |

Table 13: Hyperparameter selection for Predict&Search.

| Problem | sample-rate | learning-rate | max-epoch | patience |
|---|---|---|---|---|
| HEM_knapsack | 10 | 0.001 | 1000 | 10 |
| HEM_mis | 10 | 0.001 | 1000 | 10 |
| HEM_setcover | 10 | 0.001 | 1000 | 10 |
| vary_bounds_s1 | 10 | 0.001 | 1000 | 10 |
| vary_matrix_s1 | 10 | 0.001 | 1000 | 10 |
| vary_matrix_rhs_bounds_obj_s1 | 10 | 0.001 | 1000 | 10 |
| vary_obj_s1 | 10 | 0.001 | 1000 | 10 |
| vary_obj_s3 | 10 | 0.001 | 1000 | 10 |
| vary_rhs_s2 | 10 | 0.001 | 1000 | 10 |
| vary_rhs_s4 | 10 | 0.001 | 1000 | 10 |
| vary_rhs_obj_s2 | 10 | 0.001 | 1000 | 10 |

Table 14: Hyperparameter selection for Hybrid_Learn2Branch (Part 1).

ation metrics include objective function values and gap estimations under identical wall-clock time constraints. The complete results can be found in Tables 18, 19, 20, and 21.

In the benchmarking study, the Learn2Branch configuration was set with a maximum of 1000 epochs, and batch sizes for training, validation, and pre-training were all set to 64. The learning rate was fixed at 1e-3, and the initial sampling rate was set at 10. If effective sampling could not be achieved, the rate was progressively reduced to a minimum of 1. For the GNN&GBDT framework, we limited the solver to handle only 40% of the original problem size, aiming to highlight its effectiveness in working with smaller-scale solvers while maintaining overall solution quality.

The experimental results align with previous findings in the literature. Despite claims that machine learning (ML)-based methods can outperform traditional solvers, Gurobi remains the dominant solver across most problem instances, particularly those derived from real-world scenarios. SCIP demonstrated robustness but generally underperformed relative to Gurobi. While Large Neighborhood Search (LNS) had been reported to outperform Gurobi in certain medium- to large-scale generated problems, our experiments showed that Gurobi still outperformed LNS in many cases across a broader range of tests.

| Problem | early-stopping | epoch-size | batch-size | pretrain-batch-size | valid-batch-size |
|---|---|---|---|---|---|
| HEM_knapsack | 20 | 312 | 32 | 128 | 128 |
| HEM_mis | 20 | 312 | 32 | 128 | 128 |
| HEM_setcover | 20 | 312 | 32 | 128 | 128 |
| vary_bounds_s1 | 20 | 312 | 32 | 128 | 128 |
| vary_matrix_s1 | 20 | 312 | 32 | 128 | 128 |
| vary_matrix_rhs_bounds_obj_s1 | 20 | 312 | 32 | 128 | 128 |
| vary_obj_s1 | 20 | 312 | 32 | 128 | 128 |
| vary_obj_s3 | 20 | 312 | 32 | 128 | 128 |
| vary_rhs_s2 | 20 | 312 | 32 | 128 | 128 |
| vary_rhs_s4 | 20 | 312 | 32 | 128 | 128 |
| vary_rhs_obj_s2 | 20 | 312 | 32 | 128 | 128 |

Table 15: Hyperparameter selection for Hybrid_Learn2Branch (Part 2).

| Problem | learning-rate | max-epoch |
|---|---|---|
| HEM_knapsack | 0.003 | 200 |
| HEM_mis | 0.003 | 200 |
| HEM_setcover | 0.003 | 200 |
| vary_bounds_s1 | 0.003 | 200 |
| vary_matrix_s1 | 0.003 | 200 |
| vary_matrix_rhs_bounds_obj_s1 | 0.003 | 200 |
| vary_obj_s1 | 0.003 | 200 |
| vary_obj_s3 | 0.003 | 200 |
| vary_rhs_s2 | 0.003 | 200 |
| vary_rhs_s4 | 0.003 | 200 |
| vary_rhs_obj_s2 | 0.003 | 200 |

Table 16: Hyperparameter selection for GNN-MILP.

The Adaptive Constraint Partition Based Optimization Framework (ACP), an improvement over LNS, exhibited superior performance compared to LNS in almost all scenarios. In several cases, ACP matched or even surpassed Gurobi's performance. Among machine learning-based algorithms, the GNN&GBDT framework, which integrates small-scale solvers as sub-solvers in a large-scale integer programming framework, demonstrated excellent problem reduction capabilities and performed notably well in large-scale problems. However, due to structural limitations, GNN&GBDT is only applicable to pure integer programming problems, restricting its use in MILP scenarios.

The Learn2Branch approach, which relies on computationally expensive strong branching to collect training samples, encountered difficulties in gathering sufficient training data within reasonable time frames for many mid- to large-scale problems. In some cases, ineffective sampling resulted in errors, highlighting the need for further improvement in the sampling strategy to improve the approach's robustness across a broader set of problem instances.

In addition to the initially tested machine learning-based algorithms—Learn2Branch and GNN&GBDT—as well as traditional solvers like Gurobi, SCIP, LNS, and ACP, we evaluated four additional ML-based algorithms: Neural Diving (Nair et al., 2020b), Predict&Search (Han et al., 2023), Hybrid_Learn2Branch (Gupta et al., 2020), and GNN-MILP (Chen et al., 2022). These methods were tested on selected datasets, with results presented in Tables 20 and 21.

Neural Diving and Predict&Search showed competitive performance across certain datasets. Predict&Search demonstrated particularly strong results on small- to medium-scale problem instances, while Hybrid_Learn2Branch provided some of the best objective function values for specific problems such as vary_obj_s1. However, similar to Learn2Branch, this method encountered difficulties in certain scenarios due to challenges in collecting sufficient training data. GNN-MILP, while showing promise in synthetic problem settings, faced difficulties when applied to real-world scenarios, often producing infeasible solutions, as indicated by the ! markers in Table 20. This suggests that GNN-MILP requires further refinement to improve its generalization beyond synthetic problems.

As shown in Tables 18, 19, 20, and 21, Gurobi consistently delivered the best results across most problem instances. It performed particularly well on real-world datasets such as Coral and HEM_knapsack. SCIP and ACP performed well in certain cases but were generally outperformed by Gurobi, especially in large-scale problems where Gurobi's commercial optimization techniques excel.

Among the ML-based solvers, GNN&GBDT was competitive but had difficulty handling MIP problems, limiting its broader applicability. Learn2Branch and Hybrid_Learn2Branch showed potential, but both encountered difficulties in collecting sufficient training data for large-scale problem settings, as indicated by the + markers in Tables 18 and 19. Neural Diving and Predict&Search

| Problem | batch-size | learning-rate | num-epochs |
|---|---|---|---|
| HEM_knapsack | 1 | 0.0001 | 30 |
| HEM_mis | 1 | 0.0001 | 30 |
| HEM_setcover | 1 | 0.0001 | 30 |
| vary_bounds_s1 | 1 | 0.0001 | 30 |
| vary_matrix_s1 | 1 | 0.0001 | 30 |
| vary_matrix_rhs_bounds_obj_s1 | 1 | 0.0001 | 30 |
| vary_obj_s1 | 1 | 0.0001 | 30 |
| vary_obj_s3 | 1 | 0.0001 | 30 |
| vary_rhs_s2 | 1 | 0.0001 | 30 |
| vary_rhs_s4 | 1 | 0.0001 | 30 |
| vary_rhs_obj_s2 | 1 | 0.0001 | 30 |

Table 17: Hyperparameter selection for Neural Diving.

| | Gurobi | SCIP | LNS | ACP | Learn2branch | GNN&GBDT | Time |
|---|---|---|---|---|---|---|---|
| MIS_easy | 4598.1000 | 3723.4000 | 4587.1000 | **4610.7000** | + | 4507.5000 | 600s |
| MIS_medium | 2.18e+04 | 1.86e+04 | 2.28e+04 | **2.32e+04** | + | 2.27e+04 | 2000s |
| MIS_hard | 2.17e+05 | 9078.9000 | 2.17e+05 | 2.27e+05 | + | **2.27e+05** | 4000s |
| MVC_easy | 5383.0000 | 6291.0000 | 5395.7000 | **5368.4000** | + | 5473.3000 | 600s |
| MVC_medium | 2.82e+04 | 3.13e+04 | 2.71e+04 | **2.68e+04** | + | 2.73e+04 | 2000s |
| MVC_hard | 2.83e+05 | 4.91e+05 | 2.74e+05 | 2.76e+05 | + | **2.72e+05** | 4000s |
| SC_easy | 3301.3000 | 5047.5000 | 3252.5000 | **3190.5000** | + | 3285.6000 | 600s |
| SC_medium | 1.80e+04 | 2.52e+04 | 1.63e+04 | **1.59e+04** | + | 1.65e+04 | 2000s |
| SC_hard | 3.20e+05 | 9.19e+05 | 1.73e+05 | **1.70e+05** | + | 2.29e+05 | 4000s |
| MIPlib | **1.84e+04** | 1.84e+04 | 1.98e+04 | 1.84e+04 | 1.89e+04 | - | 150s |
| Coral | 3805.7000 | 8.48e+07 | 4.67e+08 | 1.40e+08 | + | - | 4000s |
| Cut | **2.89e+04** | 3.70e+04 | 3.35e+04 | 3.07e+04 | 3.71e+04 | - | 4000s |
| ECOGCNN | **7.56e+05** | 7.56e+05 | 7.58e+05 | 7.57e+05 | + | - | 4000s |
| HEM_knapsack | **422.6000** | **422.6000** | **422.6000** | **422.6000** | **422.6000** | 422.6000 | 100s |
| HEM_mis | **228.8000** | **228.8000** | 227.6000 | **228.8000** | **228.8000** | 216.6000 | 100s |
| HEM_setcover | **231.6000** | **231.6000** | 233.0000 | **231.6000** | **231.6000** | 231.8000 | 100s |
| HEM_corlat | **251.0000** | **251.0000** | 248.8000 | **251.0000** | ! | - | 100s |
| HEM_mik | **-6.28e+04** | **-6.28e+04** | -6.25e+04 | -6.18e+04 | ! | - | 100s |
| item_placement | **5.3000** | 10.8000 | 12.8000 | 10.7000 | 16.5000 | - | 4000s |
| load_balancing | **708.8000** | 712.0000 | 723.2000 | 709.3000 | + | - | 1000s |
| anonymous | **2.50e+05** | 1.07e+06 | 2.04e+06 | 5.29e+05 | + | - | 4000s |
| Nexp | 1.16e+08 | 1.17e+08 | 1.18e+08 | **1.16e+08** | 1.18e+08 | - | 4000s |
| Transportation | **1.24e+06** | 1.30e+06 | 1.40e+06 | 1.28e+06 | 1.31e+06 | - | 4000s |
| vary_bounds_s1 | **1.24e+04** | 1.25e+04 | 2.07e+04 | **1.24e+04** | 1.29e+04 | - | 400s |
| vary_bounds_s2 | **351.0000** | 351.0000 | 413.6000 | **351.0000** | ! | - | 1000s |
| vary_bounds_s3 | **351.0000** | 351.0000 | 417.2000 | **351.0000** | ! | - | 1000s |
| vary_matrix_s1 | **61.6000** | 62.6000 | 61.6000 | 61.6000 | 62.7000 | - | 100s |
| vary_matrix_rhs_bounds_s1 | **2.00e+09** | **2.00e+09** | 2.88e+09 | 5.89e+09 | + | - | 100s |
| vary_matrix_rhs_bounds_obj | **-5.16e+04** | -4.56e+04 | -2.13e+04 | -4.82e+04 | -4.76e+04 | - | 100s |
| vary_obj_s1 | 8625.4000 | 8630.0000 | 8642.0000 | **8625.4000** | 8633.6000 | **8625.4000** | 100s |
| vary_obj_s2 | **1169.5000** | 1171.0000 | 4045.9000 | **1169.5000** | ! | - | 150s |
| vary_obj_s3 | -2180.1000 | 30.3000 | 1127.3000 | 638.8000 | -2180.1000 | - | 100s |
| vary_rhs_s1 | **-349.5000** | -338.9000 | -54.4000 | -291.5000 | + | - | 100s |
| vary_rhs_s2 | **-1.72e+04** | -1.72e+04 | -1.67e+04 | -1.72e+04 | -1.72e+04 | - | 100s |
| vary_rhs_s3 | 5.73e+04 | 5.73e+04 | **5.73e+04** | **5.73e+04** | + | 5.73e+04 | 100s |
| vary_rhs_s4 | **-1.72e+04** | **-1.72e+04** | -1.68e+04 | -1.71e+04 | -1.72e+04 | - | 100s |
| vary_rhs_obj_s1 | **-1.79e+05** | -1.78e+05 | -1.76e+05 | -1.79e+05 | + | - | 600s |
| vary_rhs_obj_s2 | -8.08e+05 | **-8.08e+05** | -7.07e+05 | -7.64e+05 | -8.08e+05 | - | 100s |
| Aclib | **8.24e+04** | **8.24e+04** | 8.25e+04 | 8.28e+04 | ! | - | 100s |
| fc.data | **378.6000** | 378.6000 | 490.4000 | **378.6000** | ! | - | 100s |
| nn_verification | **-8.3000** | -8.4000 | -9.7000 | -9.7000 | ! | - | 100s |

Table 18: Objective function value of baselines. + represents the problem of scale being too large to accept the time to collect training samples. ! represents the problem of errors during band training. -represents MILP problems that cannot be solved by the IP framework, GNN&GBDT.

also showed promise, with Predict&Search achieving superior gap estimations on problems such as vary_matrix_s1 and vary_rhs_obj_s2.

In conclusion, while traditional solvers like Gurobi remain top performers in most cases, the inclusion of these new ML-based methods—particularly Neural Diving and Predict&Search—suggests that machine learning techniques can offer competitive performance, especially in small- to medium-scale or synthetic problem settings. However, challenges remain in scaling these methods to larger, real-world problems, and further research is necessary to overcome these limitations.

| | Gurobi | SCIP | LNS | ACP | Learn2branch | GNN&GBDT | Time |
|---|---|---|---|---|---|---|---|
| MIS_easy | 0.0908 | 0.3555 | 0.0934 | **0.0878** | + | 0.1127 | 600s |
| MIS_medium | 0.1634 | 0.3607 | 0.1077 | **0.0916** | + | 0.1148 | 2000s |
| MIS_hard | 0.1714 | 53.4844 | 0.1714 | 0.1184 | + | **0.1169** | 4000s |
| MVC_easy | 0.0751 | 0.2707 | 0.0773 | **0.0726** | + | 0.0903 | 600s |
| MVC_medium | 0.1255 | 0.2697 | 0.0895 | **0.0787** | + | 0.0976 | 2000s |
| MVC_hard | 0.1310 | 93.5867 | 0.1018 | 0.1077 | + | **0.0951** | 4000s |
| SC_easy | 0.0415 | 1.00e+20 | 0.0292 | **0.0104** | + | 0.0390 | 600s |
| SC_medium | 0.9861 | 2.00e+19 | 0.9846 | **0.9843** | + | 0.9848 | 2000s |
| SC_hard | 0.9920 | 4.25e+05 | 0.9852 | **0.9850** | + | 0.9887 | 4000s |
| MIPlib | **0.0000** | 0.0587 | 0.2157 | 0.0004 | 0.3363 | - | 150s |
| Coral | **2.85e+04** | 2.86e+19 | 3.05e+04 | 3.05e+04 | + | - | 4000s |
| Cut | **0.1490** | 0.5387 | 0.2744 | 0.1651 | 0.5782 | - | 4000s |
| ECOGCNN | **0.2512** | 4.6056 | 0.2730 | 0.2516 | + | - | 4000s |
| HEM_knapsack | **0.0000** | **0.0000** | **0.0000** | **0.0000** | **0.0000** | **0.0000** | 100s |
| HEM_mis | **0.0000** | **0.0000** | 0.0053 | **0.0000** | **0.0000** | 0.0568 | 100s |
| HEM_setcover | **0.0000** | **0.0000** | 0.0067 | **0.0000** | **0.0000** | 0.0007 | 100s |
| HEM_corlat | **0.0000** | **0.0000** | 0.0129 | **0.0000** | ! | - | 100s |
| HEM_mik | **0.0000** | **0.0000** | 0.0034 | 0.0146 | ! | - | 100s |
| item_placement | **0.6481** | 2.33e+07 | 0.8431 | 0.8076 | 4.17e+07 | - | 4000s |
| load_balancing | **0.0028** | 0.0273 | 0.0227 | 0.0035 | + | - | 1000s |
| anonymous | **0.3088** | 4.2244 | 0.9256 | 0.5449 | + | - | 4000s |
| Nexp | 0.0787 | 0.1514 | 0.1095 | **0.0754** | 0.1629 | - | 4000s |
| Transportation | **0.1512** | 0.2575 | 0.2490 | 0.1767 | 0.2725 | - | 4000s |
| vary_bounds_s1 | **0.0000** | 0.0487 | 0.3956 | **0.0000** | 0.1518 | - | 400s |
| vary_bounds_s2 | **0.0000** | 0.0864 | 0.1511 | **0.0000** | ! | - | 1000s |
| vary_bounds_s3 | **0.0000** | 0.0893 | 0.1587 | **0.0000** | ! | - | 1000s |
| vary_matrix_s1 | **0.0000** | 0.3796 | 0.0008 | 0.0008 | 0.4002 | - | 100s |
| vary_matrix_rhs_bounds_s1 | **0.0000** | **0.0000** | 0.2865 | 0.5566 | + | - | 100s |
| vary_matrix_rhs_bounds_obj | **0.0000** | 0.1128 | 1.9199 | 0.0666 | 0.0953 | - | 100s |
| vary_obj_s1 | **0.0000** | 0.0031 | 0.0019 | 0.0054 | 0.0054 | **0.0000** | 100s |
| vary_obj_s2 | **0.0000** | 6.8251 | 0.7232 | **0.0000** | ! | - | 150s |
| vary_obj_s3 | **0.0000** | 6.00e+19 | 2.1882 | 10.0836 | 0.0672 | - | 100s |
| vary_rhs_s1 | **0.0003** | 0.0364 | 5.5134 | 0.2037 | + | - | 100s |
| vary_rhs_s2 | **0.0000** | 0.0013 | 0.0274 | 0.0009 | 0.0018 | - | 100s |
| vary_rhs_s3 | 0.0001 | 0.0001 | **0.0000** | **0.0000** | + | 0.0007 | 100s |
| vary_rhs_s4 | **0.0000** | **0.0000** | 0.0224 | 0.0016 | 0.0033 | - | 100s |
| vary_rhs_obj_s1 | **0.0001** | 0.0087 | 0.0191 | 0.0039 | + | - | 600s |
| vary_rhs_obj_s2 | 0.0001 | **0.0000** | 0.1032 | 0.0367 | 0.0011 | - | 100s |
| Aclib | **0.0000** | **0.0000** | 0.0006 | 0.0028 | ! | - | 100s |
| fc.data | **0.0000** | **0.0000** | 0.1729 | **0.0000** | ! | - | 100s |
| nn_verification | **0.0001** | 0.0756 | 0.1493 | 0.1493 | ! | - | 100s |

Table 19: Gap estimation of baselines. + represents the problem of scale being too large to accept the time to collect training samples. ! represents the problem of errors during band training. -represents MILP problems that cannot be solved by the IP framework, GNN&GBDT.

| | Predict&Search | Hybrid_Learn2branch | GNN-MILP | Neural Diving | Time |
|---|---|---|---|---|---|
| HEM_knapsack | **422.6000** | 420.6000 | 0.0000 | 0.0000 | 100s |
| HEM_mis | **228.8000** | **228.8000** | 192.4000 | 0.0000 | 100s |
| HEM_setcover | **231.6000** | ! | 256.0000 | 231.6000 | 100s |
| vary_bounds_s1 | **12384.8000** | 13054.4000 | 12381.8000 | 12381.8000 | 400s |
| vary_matrix_s1 | **61.5939** | 62.9221 | ! | **61.5939** | 100s |
| vary_matrix_rhs_bounds_obj_s1 | **-51638.3000** | -27727.0100 | -51637.9000 | -51638.2900 | 100s |
| vary_obj_s1 | 8625.4000 | 8629.0000 | **8625.4000** | **8625.4000** | 100s |
| vary_obj_s3 | **-2180.0980** | ! | ! | **-2180.0980** | 100s |
| vary_rhs_s2 | **-17168.3500** | -17168.2400 | * | **-17168.3500** | 100s |
| vary_rhs_s4 | -17166.2500 | **-17166.4600** | ! | **-17166.4600** | 100s |
| vary_rhs_obj_s2 | **-807964.3000** | -807871.0000 | ! | -807962.3000 | 100s |

Table 20: Objective function value of new added baseline on selected datasets. ! represents the problem of errors during training. * represents that can not find a feasible solution.

| | Predict&Search | Hybrid_Learn2branch | GNN-MILP | Neural Diving | Time |
|---|---|---|---|---|---|
| HEM_knapsack | **0.0000** | 0.0047 | $+\infty$ | $+\infty$ | 100s |
| HEM_mis | **0.0000** | **0.0000** | 0.1908 | $+\infty$ | 100s |
| HEM_setcover | **0.0000** | ! | 0.0404 | 0.0976 | 100s |
| vary_bounds_s1 | **0.0179** | 0.1430 | 0.0363 | 0.0363 | 400s |
| vary_matrix_s1 | **0.0000** | 0.5513 | ! | **0.0000** | 100s |
| vary_matrix_rhs_bounds_obj_s1 | **3.81e-05** | 2.0e19 | 0.0048 | 3.91e-05 | 100s |
| vary_obj_s1 | **0.0000** | 0.0035 | 0.0025 | 0.0009 | 100s |
| vary_obj_s3 | **0.0000** | ! | ! | 0.1275 | 100s |
| vary_rhs_s2 | **1.3e-5** | 0.0001 | * | 1.6e-5 | 100s |
| vary_rhs_s4 | 3.7e-5 | **0.0000** | ! | 5.4e-5 | 100s |
| vary_rhs_obj_s2 | **6.9e-5** | 0.0011 | ! | 1.6e-5 | 100s |

Table 21: Gap estimation of new added baseline on selected datasets. ! represents the problem of errors during training. * represents that can not find a feasible solution.

| | avg_obj | obj_std_error | obj_error_bar | avg_gap | gap_std_error | gap_error_bar |
|---|---|---|---|---|---|---|
| MIS_easy | 4.36e+03 | 18.4347 | 30.6534 | 0.1506 | 0.0020 | 0.0026 |
| MIS_medium | 2.32e+04 | 32.1560 | 50.5459 | 0.0923 | 0.0020 | 0.0033 |
| MIS_hard | + | + | + | + | + | + |
| MVC_easy | 5.61e+03 | 28.9665 | 54.5941 | 0.1130 | 0.0043 | 0.0068 |
| MVC_medium | + | + | + | + | + | + |
| MVC_hard | + | + | + | + | + | + |
| SC_easy | 3.23e+03 | 18.6588 | 33.5314 | 0.0224 | 0.0047 | 0.0091 |
| SC_medium | + | + | + | + | + | + |
| SC_hard | + | + | + | + | + | + |
| MIPlib | 1.84e+04 | 3.23e+04 | 5.59e+04 | 0.0000 | 0.0000 | 0.0000 |
| Coral | 14.5999 | 22.8699 | 45.3997 | 2.33e+04 | 4.66e+04 | 9.32e+04 |
| Cut | 2.93e+04 | 1.98e+04 | 2.80e+04 | 0.1568 | 0.1775 | 0.2491 |
| ECOGCNN | 7.56e+05 | 1.07e+06 | 1.51e+06 | 0.2512 | 0.3527 | 0.4988 |
| HEM_knapsack | 422.6000 | 12.7844 | 22.6000 | 0.0000 | 0.0000 | 0.0000 |
| HEM_mis | 228.8000 | 3.6551 | 6.8000 | 0.0000 | 0.0000 | 0.0000 |
| HEM_setcover | 231.6000 | 28.5909 | 47.4000 | 0.0000 | 0.0000 | 0.0000 |
| HEM_corlat | 251.0000 | 135.0704 | 269.0000 | 0.0000 | 0.0000 | 0.0000 |
| HEM_mik | -6.28e+04 | 1.46e+04 | 2.91e+04 | 0.0000 | 0.0000 | 0.0000 |
| item_placement | 5.5310 | 1.0528 | 2.7063 | 0.6595 | 0.1639 | 0.3395 |
| load_balancing | 708.8000 | 21.7246 | 41.2000 | 0.0028 | 0.0001 | 0.0002 |
| anonymous | 2.46e+05 | 1.67e+05 | 2.77e+05 | 0.2909 | 0.1246 | 0.1874 |
| Nexp | 1.16e+08 | 2.10e+08 | 4.18e+08 | 0.0759 | 0.1345 | 0.2679 |
| Transportation | 1.25e+06 | 1.32e+04 | 2.51e+04 | 0.1568 | 0.0050 | 0.0084 |
| vary_bounds_s1 | 1.24e+04 | 863.5863 | 1.13e+03 | 0.0003 | 0.0005 | 0.0010 |
| vary_bounds_s2 | 355.0000 | 0.0000 | 0.0000 | 0.0113 | 0.0000 | 0.0000 |
| vary_bounds_s3 | 355.0000 | 0.0000 | 0.0000 | 0.0113 | 0.0000 | 0.0000 |
| vary_matrix_s1 | 61.5939 | 0.9036 | 1.5450 | 0.0000 | 0.0000 | 0.0000 |
| vary_matrix_rhs_bounds_s1 | 2.00e+09 | 1.31e+08 | 2.11e+08 | 0.0000 | 0.0000 | 0.0001 |
| vary_matrix_rhs_bounds_obj_s1 | -5.16e+04 | 2.16e+04 | 3.46e+04 | 0.0000 | 0.0000 | 0.0001 |
| vary_obj_s1 | 8.63e+03 | 101.7342 | 184.6000 | 0.0000 | 0.0000 | 0.0000 |
| vary_obj_s2 | 1.17e+03 | 1.12e+03 | 2.12e+03 | 0.0000 | 0.0000 | 0.0000 |
| vary_obj_s3 | -2.18e+03 | 352.3199 | 598.5451 | 0.0000 | 0.0000 | 0.0000 |
| varh_rhs_s1 | -349.4640 | 5.8607 | 10.2160 | 0.0003 | 0.0005 | 0.0011 |
| vary_rhs_s2 | -1.72e+04 | 1.6904 | 3.3795 | 0.0000 | 0.0000 | 0.0000 |
| vary_rhs_s3 | 5.73e+04 | 6.14e+04 | 1.11e+04 | 0.0001 | 0.0000 | 0.0000 |
| vary_rhs_s4 | -1.72e+04 | 3.0168 | 3.7753 | 0.0000 | 0.0000 | 0.0000 |
| vary_rhs_obj_s1 | -1.79e+05 | 4.09e+04 | 6.33e+04 | 0.0001 | 0.0000 | 0.0000 |
| vary_rhs_obj_s2 | -8.08e+05 | 3.53e+05 | 6.94e+05 | 0.0001 | 0.0000 | 0.0000 |
| Aclib | 8.24e+04 | 9.40e+04 | 2.31e+05 | 0.0000 | 0.0000 | 0.0000 |
| fc.data | 378.6000 | 196.6750 | 384.4000 | 0.0000 | 0.0000 | 0.0000 |
| nn_verification | -8.2514 | 9.6160 | 24.5807 | 0.0001 | 0.0000 | 0.0001 |

Table 22: Experimental results of Predict&Search on selected datasets. + represents the problem of scale being too large to accept the time to collect training samples.

| | Gurobi | SCIP | LNS | ACP | Learn2branch | GNN&GBDT |
|---|---|---|---|---|---|---|
| MIS_easy | 36.7445 | 49.5063 | 34.6024 | 34.1673 | + | 21.4168 |
| MIS_medium | 30.6104 | 82.5639 | 59.6125 | 54.2726 | + | 57.9730 |
| MIS_hard | 197.1507 | 159.7765 | 222.3543 | 1648.7565 | + | 273.1787 |
| MCV_easy | 21.3176 | 26.2868 | 18.2733 | 27.6136 | + | 27.3013 |
| MVC_medium | 101.2970 | 170.3988 | 113.3693 | 112.3249 | + | 116.6010 |
| MVC_hard | 303.5272 | 263.6697 | 262.7302 | 1719.3960 | + | 647.3696 |
| SC_easy | 326.5255 | 31.3076 | 29.4139 | 23.8124 | + | 30.7511 |
| SC_medium | 99.3993 | 174.7215 | 82.8357 | 79.5404 | + | 66.2896 |
| SC_hard | 493.9357 | 1122.9085 | 1415.5947 | 6466.7330 | + | 3.90e+04 |
| MIPlib | 5.59e+04 | 5.59e+04 | 5.99e+04 | 5.59e+04 | 5.73e+04 | - |
| Coral | 2.28e+04 | 5.09e+08 | 2.80e+09 | 8.40e+08 | + | - |
| Cut | 2.75e+04 | 4.29e+04 | 3.09e+04 | 3.10e+04 | 4.31e+04 | - |
| ECOGCNN | 1.51e+06 | 1.51e+06 | 1.52e+06 | 1.51e+06 | + | - |
| HEM_knapsack | 22.6000 | 22.6000 | 22.6000 | 22.6000 | 22.6000 | 22.6000 |
| HEM_mis | 6.8000 | 6.8000 | 7.6000 | 6.8000 | 6.8000 | 10.6000 |
| HEM_setcover | 47.4000 | 47.4000 | 46.0000 | 47.4000 | 47.4000 | 48.2000 |
| HEM_corlat | 269.0000 | 269.0000 | 271.2000 | 269.0000 | ! | - |
| HEM_mik | 2.91e+04 | 2.91e+04 | 2.89e+04 | 2.81e+04 | ! | - |
| item_placement | 2.5066 | 3.9976 | 5.9915 | 6.4424 | 5.2309 | - |
| load_balancing | 41.2000 | 42.0000 | 40.2000 | 41.7000 | + | - |
| anonymous | 2.59e+05 | 1.95e+06 | 1.11e+06 | 6.77e+05 | + | - |
| Nexp | 4.18e+08 | 4.19e+08 | 4.22e+08 | 4.18e+08 | 4.22e+08 | - |
| Transportation | 2.23e+04 | 2.61e+04 | 4.90e+04 | 2.94e+04 | 2.15e+04 | - |
| vary_bounds_s1 | 1130.2000 | 1256.8000 | 2502.6000 | 1130.2000 | 1333.6000 | - |
| vary_bound_s2 | 0.0000 | 0.0000 | 14.6000 | 0.0000 | ! | - |
| vary_bounds_s3 | 0.0000 | 0.0000 | 0.8000 | 0.0000 | ! | - |
| vary_matrix_s1 | 1.5450 | 0.7618 | 1.4971 | 1.5403 | 1.6728 | - |
| vary_matrix_rhs_bounds_s1 | 2.11e+08 | 2.11e+08 | 1.18e+09 | 7.99e+09 | + | - |
| vary_matrix_rhs_bounds_obj | 3.46e+04 | 2.85e+04 | 2.04e+04 | 3.26e+04 | 3.08e+04 | - |
| vary_obj_s1 | 184.6000 | 181.0000 | 190.0000 | 184.6000 | 186.4000 | 184.6000 |
| vary_obj_s2 | 2120.3583 | 2118.8228 | 3292.1402 | 2120.3530 | ! | - |
| vary_obj_s3 | 598.5451 | 650.1423 | 6178.0536 | 6588.9847 | 598.5451 | - |
| vary_rhs_s1 | 10.2160 | 28.3040 | 12.8240 | 34.0080 | + | - |
| vary_rhs_s2 | 3.3795 | 1.1493 | 74.5091 | 13.0051 | 2.2173 | - |
| vary_rhs_s3 | 2.19e+04 | 2.19e+04 | 2.19e+04 | 2.19e+04 | + | 2.18e+04 |
| vary_rhs_s4 | 3.9644 | 3.9789 | 278.7842 | 26.1177 | 5.1057 | - |
| vary_rhs_obj_s1 | 6.33e+04 | 6.39e+04 | 6.05e+04 | 6.35e+04 | + | - |
| vary_rhs_obj_s2 | 6.94e+05 | 6.94e+05 | 3.53e+05 | 5.37e+05 | 6.94e+05 | - |
| Aclib | 1.78e+05 | 1.78e+05 | 1.78e+05 | 1.80e+05 | ! | - |
| fc.data | 384.4000 | 384.4000 | 307.6000 | 384.4000 | ! | - |
| nn_verification | 24.5807 | 24.7198 | 24.5584 | 24.5584 | ! | - |

Table 23: The error bar of objective function value. + represents the problem of scale being too large to accept the time to collect training samples. ! represents the problem of errors during band training. -represents MILP problems that cannot be solved by the IP framework, GNN&GBDT.

## C.6 ERROR BARS OF BENCHMARKING STUDY

To enhance the reliability and reproducibility of our benchmarking study, we analyzed the error bars from multiple experiments. The results are displayed in Tables 23 and Table 24, respectively showing the error bars for objective function values and gap estimates. Under various data conditions, the solvers Gurobi and SCIP and the machine learning-based optimization algorithms Learn2branch and GNN&GBDT exhibited consistent stability across different problems.

However, the classical optimization algorithms LNS and ACP demonstrated significant instability in some instances, such as with the datasets "vary_obj_s3" and "Coral." This instability can likely be attributed to these algorithms' heavy reliance on selecting initial feasible solutions. In complex problem spaces, the distribution of initial feasible solutions may exhibit randomness, leading to instability in the final solutions. This aspect underlines the importance of considering initial solution strategies and their impact on the performance of optimization algorithms, particularly in diverse and challenging problem settings.

## C.7 STANDARD DEVIATIONS OF BENCHMARKING STUDY

In benchmark studies, the stability of baseline methods is often assessed using metrics such as error bars and standard deviations. In this work, we provide additional insights into the stability of the methods by reporting standard deviations alongside the objective values and gap estimations. Specifically, Tables 23 and 24 in the appendix present the error bars for both the objective and gap values across various problem instances.

| | Gurobi | SCIP | LNS | ACP | Learn2branch | GNN&GBDT |
|---|---|---|---|---|---|---|
| MIS_easy | 0.0081 | 0.0131 | 0.0076 | 0.0075 | + | 0.0047 |
| MIS_medium | 0.0014 | 0.0044 | 0.0026 | 0.0023 | + | 0.0026 |
| MIS_hard | 0.0009 | 0.0179 | 0.0010 | 0.0072 | + | 0.0012 |
| MCV_easy | 0.0039 | 0.0042 | 0.0034 | 0.0051 | + | 0.0050 |
| MVC_medium | 0.0036 | 0.0055 | 0.0042 | 0.0042 | + | 0.0042 |
| MVC_hard | 0.0011 | 0.0005 | 0.0010 | 0.0063 | + | 0.0024 |
| SC_easy | 0.0900 | 0.0062 | 0.0091 | 0.0075 | + | 0.0094 |
| SC_medium | 0.0055 | 0.0070 | 0.0051 | 0.0050 | + | 0.0040 |
| SC_hard | 0.0015 | 0.0012 | 0.0083 | 0.0395 | + | 0.1456 |
| MIPlib | 2047.4498 | 1842.6534 | 1410.9864 | 2047.4730 | 1350.1054 | - |
| Coral | 1.90e+08 | 2.01e+22 | 2.49e+13 | 7.47e+12 | + | - |
| Cut | 1.0088 | 1.5437 | 1.0304 | 1.1291 | 1.5354 | - |
| ECOGCNN | 2.70e+04 | 2.10e+04 | 2.11e+04 | 2.70e+04 | + | - |
| HEM_knapsack | 0.0565 | 0.0565 | 0.0565 | 0.0565 | 0.0565 | 0.0565 |
| HEM_mis | 0.0306 | 0.0306 | 0.0345 | 0.0306 | 0.0306 | 0.0515 |
| HEM_setcover | 0.2254 | 0.2254 | 0.2073 | 0.2254 | 0.2254 | 0.2265 |
| HEM_corlat | 0.5305 | 0.5305 | 0.5215 | 0.5305 | ! | - |
| HEM_mik | 0.8654 | 0.8654 | 0.8584 | 0.8357 | ! | - |
| item_placement | 0.8876 | 0.5874 | 0.4809 | 0.5202 | 0.4637 | - |
| load_balancing | 0.0611 | 0.0579 | 0.0589 | 0.0602 | + | - |
| anonymous | 2.1649 | 8.1505 | 0.8448 | 6.0986 | + | - |
| Nexp | 1.85e+06 | 1.85e+06 | 2.00e+06 | 1.85e+06 | 1.90e+06 | - |
| Transportation | 0.0183 | 0.0205 | 0.0362 | 0.0235 | 0.0167 | - |
| vary_bounds_s1 | 0.0836 | 0.0911 | 0.1080 | 0.0836 | 0.0971 | - |
| vary_bound_s2 | 0.0000 | 0.0000 | 0.0366 | 0.0000 | ! | - |
| vary_bounds_s3 | 0.0000 | 0.0000 | 0.0019 | 0.0000 | ! | - |
| vary_matrix_s1 | 0.0245 | 0.0123 | 0.0237 | 0.0244 | 0.0260 | - |
| vary_matrix_rhs_bounds_s1 | 0.1181 | 0.1181 | 0.2905 | 0.6868 | + | - |
| vary_matrix_rhs_bounds_obj | 2.0235 | 1.6706 | 2.8904 | 2.0977 | 1.8306 | - |
| vary_obj_s1 | 0.0210 | 0.0205 | 0.0215 | 0.0210 | 0.0211 | 0.0210 |
| vary_obj_s2 | 4.0368 | 4.0243 | 4.3674 | 4.0368 | ! | - |
| vary_obj_s3 | 0.3042 | 1.0641 | 5.4783 | 10.9865 | 0.3042 | - |
| vary_rhs_s1 | 0.0284 | 0.0911 | 0.1954 | 0.1321 | + | - |
| vary_rhs_s2 | 0.0002 | 0.0001 | 0.0044 | 0.0008 | 0.0001 | - |
| vary_rhs_s3 | 0.6191 | 0.6189 | 0.6190 | 0.6191 | + | 0.6162 |
| vary_rhs_s4 | 0.0002 | 0.0002 | 0.0169 | 0.0015 | 0.0003 | - |
| vary_rhs_obj_s1 | 0.3073 | 0.3076 | 0.2931 | 0.3020 | + | - |
| vary_rhs_obj_s2 | 0.4622 | 0.4622 | 0.3328 | 0.4127 | 0.4622 | - |
| Aclib | 5.9017 | 5.9017 | 5.9037 | 5.9271 | ! | - |
| fc.data | 0.8559 | 0.8559 | 1.3806 | 0.8559 | ! | - |
| nn_verification | 3.0632 | 3.1317 | 1.6548 | 1.6548 | ! | - |

Table 24: The error bar of gap estimation. + represents the problem of scale being too large to accept the time to collect training samples. ! represents the problem of errors during band training. -represents MILP problems that cannot be solved by the IP framework, GNN&GBDT.

| | Gurobi | SCIP | LNS | ACP | Learn2branch | GNNGBDT |
|---|---|---|---|---|---|---|
| MIS_easy | 21.5005 | 29.0341 | 20.4964 | 20.2003 | + | 16.0027 |
| MIS_medium | 19.3987 | 58.5580 | 37.6426 | 33.0777 | + | 39.6546 |
| MIS_hard | 120.8925 | 97.5889 | 124.2721 | 909.1139 | + | 167.5108 |
| MVC_easy | 13.8075 | 18.7708 | 13.6796 | 16.1958 | + | 17.6541 |
| MVC_medium | 62.4095 | 130.4566 | 69.8594 | 68.5705 | + | 73.5231 |
| MVC_hard | 205.6697 | 171.6232 | 167.2595 | 1045.0330 | + | 347.4271 |
| SC_easy | 164.1662 | 19.9643 | 20.3551 | 14.40201 | + | 16.6546 |
| SC_medium | 61.0949 | 106.7979 | 58.4282 | 49.2778 | + | 46.6850 |
| SC_hard | 293.1404 | 609.7831 | 971.0688 | 3344.7140 | + | 26217.9500 |
| MIPlib | 32273.2986 | 32273.0200 | 34572.4900 | 32273.1800 | 33097.8200 | - |
| Coral | 9288.9841 | 2.08e+08 | 1.14e+09 | 3.43e+08 | + | - |
| Cut | 19438.9173 | 30348.1100 | 21880.4300 | 21924.4400 | 30492.4700 | - |
| ECOGCNN | 1068850.9350 | 1068849.0 | 1072247.0 | 1070354.0 | + | - |
| HEM_knapsack | 12.7844 | 12.7844 | 12.7844 | 12.7844 | 12.7844 | 12.7844 |
| HEM_mis | 3.6551 | 3.6551 | 3.9294 | 3.6551 | 3.6551 | 6.2801 |
| HEM_setcover | 28.5909 | 28.5909 | 27.3934 | 28.5909 | 28.5909 | 28.9234 |
| HEM_corlat | 135.0704 | 135.0704 | 136.2606 | 135.0704 | ! | - |
| HEM_mik | 14597.0829 | 14597.0800 | 14488.5800 | 14094.3000 | ! | - |
| item_placement | 1.1097 | 2.4419 | 3.0860 | 2.7816 | 2.7569 | - |
| load_balancing | 21.7246 | 21.4103 | 20.6630 | 21.6936 | + | - |
| anonymous | 159911.2884 | 1142112 | 727716.3 | 419905.2 | + | - |
| Nexp | 2.1e+08 | 2.1e+08 | 2.12e+08 | 2.1e+08 | 2.12e+08 | - |
| Transportation | 11884.7873 | 15847.25 | 36827.44 | 20618.52 | 13269.68 | - |
| vary_bounds_s1 | 865.9897 | 865.157 | 1436.246 | 865.9897 | 1068.365 | - |
| vary_bounds_s2 | 0.0000 | 0.0000 | 7.3103 | 0.0000 | ! | - |
| vary_bounds_s3 | 0.0000 | 0.0000 | 0.4000 | 5.22e-07 | ! | - |
| vary_matrix_s1 | 0.9036 | 0.5915 | 0.8855 | 0.8996 | 1.2523 | - |
| vary_matrix_rhs_bounds_s1 | 1.31e+08 | 1.31e+08 | 5.94e+08 | 4e+09 | + | - |
| vary_matrix_rhs_bounds_obj_s1 | 21607.4416 | 14560.27 | 13261.15 | 18427.69 | 19695 | - |
| vary_obj_s1 | 101.7342 | 100.5107 | 104.858 | 101.7342 | 102.2518 | 101.7342 |
| vary_obj_s2 | 1120.8703 | 1120.516 | 2426.985 | 1120.868 | ! | - |
| vary_obj_s3 | 352.3199 | 445.152 | 3413 | 3398.318 | 352.3199 | - |
| vary_rhs_s1 | 5.8607 | 14.4301 | 7.0876 | 19.3488 | + | - |
| vary_rhs_s2 | 1.6904 | 0.5765 | 56.2404 | 7.6243 | 1.1923 | - |
| vary_rhs_s3 | 11134.2517 | 11131.88 | 11132.79 | 11133.84 | + | 11107.51 |
| vary_rhs_s4 | 3.0156 | 3.0021 | 148.6948 | 17.5912 | 3.5953 | - |
| vary_rhs_obj_s1 | 40944.9046 | 40723.93 | 38629.9 | 40612.03 | + | - |
| vary_rhs_obj_s2 | 353075.129 | 353120.6 | 187573.9 | 276288.8 | 353085.4 | - |
| Aclib | 7822.2633 | 7822.263 | 7872.819 | 7885.637 | ! | - |
| fc.data | 196.6750 | 196.675 | 242.2813 | 196.675 | ! | - |
| nn_verification | 9.6159 | 9.6724 | 9.4476 | 9.4476 | ! | - |

Table 25: The standard deviations of objective function value.

To further enhance the robustness of our analysis, we have now included the standard deviations of the objective values and gap values for different baseline methods across various problems. These can be found in Tables 25 and 26 of the supplementary material. The inclusion of these standard deviation values allows for a clearer understanding of the variability in performance across different methods and problem classes, offering a more detailed perspective on the stability of the baseline methods.

## D    RELATED WORK

### D.1    MIXED INTEGER LINEAR PROGRAMMING

Mixed Integer Linear Programming (MILP) problem is a significant class within combinatorial optimization problems. With advancements in theoretical techniques and commercial solvers, MILP has become a fundamental problem type for modeling and solving practical issues across various fields. Formally, a MILP problem can be represented as follows:

$$
\min_x c^T x,
$$
$$
\text{subject to } Ax \leq b,
$$
$$
l \leq x \leq u,
$$
$$
x_i \in \mathbb{Z},
$$
$$
i \in \mathbb{I},
$$
(12)

where $x$ represents the decision variables, with dimension denoted by $n \in \mathbb{Z}$, and $l, u, c \in \mathbb{R}^n$ correspond to the lower bounds, upper bounds, and coefficient values of the decision variables, respectively. The matrix $A \in \mathbb{R}^{m \times n}$ and the vector $b \in \mathbb{R}^m$ define the linear constraints of the problem. The set $\mathbb{I} \subseteq \{1, 2, \ldots, n\}$ denotes the indices of variables constrained to be integer values. A solution to the MILP is considered feasible if the decision variable vector $x \in \mathbb{R}^n$ satisfies all

|  | Gurobi | SCIP | LNS | ACP | Learn2branch | GNNGBDT |
|---|---|---|---|---|---|---|
| MIS_easy | 0.0013 | 0.0073 | 0.0020 | 0.0016 | + | 0.0030 |
| MIS_medium | 0.0008 | 0.0051 | 0.0016 | 0.0020 | + | 0.0025 |
| MIS_hard | 0.0007 | 0.6056 | 0.0006 | 0.0052 | + | 0.0012 |
| MVC_easy | 0.0012 | 0.0023 | 0.0008 | 0.0015 | + | 0.0016 |
| MVC_medium | 0.0006 | 0.0043 | 0.0010 | 0.0010 | + | 0.0011 |
| MVC_hard | 0.0005 | 0.9541 | 0.0003 | 0.0032 | + | 0.0009 |
| SC_easy | 0.0428 | 0.0000 | 0.0025 | 0.0008 | + | 0.0017 |
| SC_medium | 0.0003 | 4e+19 | 0.0003 | 0.0003 | + | 0.0003 |
| SC_hard | 5.26e-05 | 110596.4 | 8.79e-05 | 0.0003 | + | 0.0013 |
| MIPlib | 4.28e-05 | 0.0526 | 0.1853 | 0.0006 | 0.2812 | - |
| Coral | 69736.34 | 4.52e+19 | 74629.05 | 74629.13 | + | - |
| Cut | 0.1765 | 0.5917 | 0.1451 | 0.2011 | 0.6116 | - |
| ECOGCNN | 0.3527 | 6.4973 | 0.3766 | 0.3523 | + | - |
| HEM_knapsack | 0.0000 | 0.0000 | 0.0000 | 0.0000 | 0.0000 | 0.0000 |
| HEM_mis | 0.0000 | 0.0000 | 0.0043 | 0.0000 | 0.0000 | 0.0184 |
| HEM_setcover | 0.0000 | 0.0000 | 0.0086 | 0.0000 | 0.0000 | 0.0014 |
| HEM_corlat | 0.0000 | 0.0000 | 0.0257 | 0.0000 | ! | - |
| HEM_mik | 6.16e-15 | 0.0000 | 0.0068 | 0.0127 | ! | - |
| item_placement | 0.1617 | 63245483 | 0.0818 | 0.0879 | 92385906 | - |
| load_balancing | 8.65e-05 | 0.0089 | 0.0041 | 0.0007 | + | - |
| anonymous | 0.0920 | 3.5603 | 0.0180 | 0.2357 | + | - |
| Nexp | 0.1402 | 0.2481 | 0.1337 | 0.1318 | 0.2463 | - |
| Transportation | 0.0006 | 0.0039 | 0.0174 | 0.0121 | 0.0078 | - |
| vary_bounds_s1 | 3.46e-05 | 0.0122 | 0.0793 | 3.46e-05 | 0.0359 | - |
| vary_bounds_s2 | 0.0000 | 0.0024 | 0.0154 | 0.0000 | ! | - |
| vary_bounds_s3 | 0.0000 | 0.0000 | 0.0008 | 1.49e-09 | ! | - |
| vary_matrix_s1 | 0.0000 | 0.0698 | 0.0016 | 0.0008 | 0.0297 | - |
| vary_matrix_rhs_bounds_s1 | 3.15r-05 | 5.41e-06 | 0.0881 | 0.1555 | + | - |
| vary_matrix_rhs_bounds_obj_s1 | 4.48e-05 | 0.2003 | 0.9437 | 0.0673 | 0.1093 | - |
| vary_obj_s1 | 0.0000 | 0.0008 | 0.0005 | 0.0000 | 0.0009 | 0.0000 |
| vary_obj_s2 | 3.93e-07 | 8.1891 | 0.1437 | 7.59e-07 | ! | - |
| vary_obj_s3 | 0.0000 | 4.9e+19 | 2.7284 | 15.9961 | 0.1344 | - |
| vary_rhs_s1 | 0.0005 | 0.0664 | 0.6963 | 0.0710 | + | - |
| vary_rhs_s2 | 1.73e-05 | 0.0004 | 0.0035 | 0.0005 | 0.0002 | - |
| vary_rhs_s3 | 1.77e-05 | 5.62e-05 | 2.36e-05 | 8.8e-06 | + | 0.0008 |
| vary_rhs_s4 | 3.13e-05 | 0.0000 | 0.0092 | 0.0010 | 9.11e-05 | - |
| vary_rhs_obj_s1 | 1.33e-05 | 0.0056 | 0.0098 | 0.0043 | + | - |
| vary_rhs_obj_s2 | 1.95e-05 | 0.0000 | 0.1574 | 0.0595 | 0.0005 | - |
| Aclib | 8.97e-06 | 0.0000 | 0.0024 | 0.0031 | ! | - |
| fc.data | 0.0000 | 0.0000 | 0.2091 | 1.3e-08 | ! | - |
| nn_verification | 3.08e-05 | 0.2137 | 0.1577 | 0.1577 | ! | - |

Table 26: The standard deviations of gap estimation.

the constraints specified in Equation (12). Among feasible solutions, the one that minimizes the objective function value is deemed optimal (Schrijver, 1998).

Based on the formulation of MILP, Gasse's proposed MILP bipartite graph representation (Gasse et al., 2019) achieves a lossless translation of the MILP problem into a graph format, serving as input for the neural embedding network (Nair et al., 2020b). As shown in Figure 1, the $n$ decision variables in MILP are represented as the set of variable nodes on the right side of the bipartite graph, while the $m$ linear constraints are represented as the set of constraint nodes on the left side. An edge connecting a variable node and a constraint node signifies the presence of the corresponding variable in that constraint.

Furthermore, people often need to solve a series of homogeneous MILP problems in real-world scenarios. By "homogeneous" (Yang et al., 2023), we mean that the generated MILP problems correspond to the same mathematical model. For example, the minimum vertex cover problem on a graph with 200 vertices and the same problem on a graph with 500 vertices are homogeneous. In contrast, the minimum vertex cover problem and the maximum cut problem are heterogeneous. Homogeneous problems share similar structures, providing an opportunity to use machine learning methods to learn the mapping from problem structures to key information for solving these problems.

## D.2 SCORE METRIC

To summarize the results in Tables 2 and 3, we provided a score metric based on gap estimation, which we apologize for not clearly defining in the main text. We appreciate the reviewer's attention to this detail. Specifically, to convert the calculated gap values into a 0–100 score, we experimented with various mapping functions, including the Sigmoid function, and ultimately chose the normal distribution function for scoring. Specifically, we set $\text{Score}(x) \sim \frac{1}{\sqrt{2\pi}\sigma} e^{-\frac{(x-\mu)^2}{\sigma^2}}$, where $x$ represents the gap estimation of the baseline method on the corresponding problem.

The rationale behind selecting the normal distribution function lies in its alignment with the law of large numbers, which suggests that as the sample size increases, the sample mean will converge to the expected value. Therefore, using a normal distribution for scoring may better reflect the natural distribution of the data, providing a more accurate representation of the performance of different baselines. Additionally, the central limit theorem indicates that the sum of multiple independent random variables, regardless of their initial distribution, will tend to follow a normal distribution as the sample size approaches infinity. This implies that even if the initial distribution of the gap data is not normal, the distribution of gaps across multiple instances may tend towards normality. Consequently, scoring based on a normal distribution can capture this underlying statistical property, thereby providing a more objective assessment of baseline performance.

Moreover, the probability density function (PDF) of the normal distribution has a natural scaling property, with sharper changes near the center and more gradual changes in the tails. By scaling the PDF values to a 0–100 range, this property allows for a more nuanced evaluation of gap sizes, enhancing the sensitivity and distinctiveness of the scores. Considering that solutions with a gap greater than 200% are typically unusable, we ultimately chose a normal distribution centered at zero with a standard deviation of 0.5 as our scoring function.

