# OpenReview forum: "ML4MILP: A Benchmark Dataset for Machine Learning-based Mixed-Integer Linear Programming"
_ICLR.cc/2025/Conference — ICLR 2025 Conference Withdrawn Submission_

### Official Review · Reviewer_L2MG · 2024-11-01

**Soundness:** 3
**Presentation:** 2
**Contribution:** 3
**Rating:** 6
**Confidence:** 3

**Summary:**

The paper introduces ML4MILP, an open-source benchmark dataset for machine learning-based approaches to solving MILP problems, featuring 100,000 instances across over 70 heterogeneous classes. To address the lack of homogeneity in existing MILP datasets, the authors developed novel structural and embedding similarity metrics, along with a new classification method to refine the categorization of less homogeneous datasets. Additionally, the paper includes comprehensive testing to establish baselines for effective benchmarking.

**Strengths:**

1. ML4MILP offers an extensive benchmark dataset with 100,000 MILP instances in over 70 classes.

2. The paper introduces a novel classification method using structural and neural embedding similarity metrics to enhance the homogeneity of existing MILP datasets.

3. A comprehensive baseline library is provided, facilitating consistent benchmarking and clear comparison across various optimization algorithms.

**Weaknesses:**

1. There could be potential in combining the structural and neural embedding metrics into a unified measure. Currently, these metrics seem to be more qualitative than quantitative, as shown in Figure 5 where cluster 6 has large distances in both metrics, but the other clusters show discrepancies (e.g., cluster 1 has a higher structural distance than cluster 3, yet a lower neural embedding distance). Merging these metrics might help address such inconsistencies and provide more comprehensive evaluations.

2. It would be valuable to identify classes that are neither too similar nor too different to facilitate benchmarking for transfer learning or other generalization approaches, which would broaden the dataset’s applicability.

3. In the baseline algorithm comparison, the quantified scores only consider solution gaps; including solving time as an additional metric would provide a more holistic assessment of algorithm performance.

**Questions:**

1. Building on weakness 1, Figure 5 shows that many inner-cluster distances are larger than distances between clusters, which raises questions about the effectiveness of the current metrics in capturing true internal homogeneity.

2. It is unclear from the provided details whether the y-axis in Figures 6 and 7 represents change values (relative differences) or absolute loss values.

---

### Official Review · Reviewer_nckF · 2024-11-02

**Soundness:** 2
**Presentation:** 3
**Contribution:** 2
**Rating:** 3
**Confidence:** 4

**Summary:**

The authors present ML4MILP, a benchmarking dataset of 100,000 instances across 70+ classes. They have baselined this library and used GNN to classify the instances, using a statistical structural similarity metric.

**Strengths:**

The paper presents a review of the state-of-the-art in MILP benchmarks and highlights the shortcomings.
The dataset is provided in three parts, including a baseline library that is benchmarked on SoTA solvers and ML algorithms. The pros and cons of these aspects are well discussed.

**Weaknesses:**

The authors have used spectral clustering with euclidean distance as a measure for classification of the dataset instances. This choice of this approach needs to be substantiated better. Specifically the choice and implementation of the clustering method, including the choice of parameters and number of clusters needs to be provided. A discussion on the scalability of this method is also required.

The use of the phrase "ML-based algorithms" should be clarified. This is a very broad term and could be seen in multiple contexts based on the reader's domain of expertise. As such, it may be better to call out the scope clearly.

Details on the specific steps for computation of structural and neural embeddings should also be included. For example, is there a reference for (2), what happens in limiting cases? What is |I| ?

It is not clear how reclassification of the dataset leads to addressing issues such as loss leading to NaN values. More analysis is required to substantiate such a claim.

The repository link provided is not accessible, it returns a connection timeout error.

Limitations in the current and possible (specific) directions for future work should be included.

Some other aspects of the paper:
The graph (Figs 6 and 7) legend has a typo for 'befor'. The horizontal axis labels are also missing. What is shown on the Y-axis - change in loss function values or absolute values? How do we interpret this?
The tables could be made concise, especially when lot of the values in the tables are repetitive (Tables 10,11,12 for eg.)

**Questions:**

a. Provide details on the choice of the classification approach.
b. Include (clarify) details for ML-based algorithms.
c. Include details for embedding computation.
d. Fix and address issues related to loss plots and associated details.
e. Please check the repository access.

**Details Of Ethics Concerns:**

Not applicable.

---

> ### Comment · Reviewer_nckF · 2024-11-27
>
> I retain my score since no rebuttal has been provided.

---

### Official Review · Reviewer_sUsa · 2024-11-03

**Soundness:** 2
**Presentation:** 2
**Contribution:** 2
**Rating:** 3
**Confidence:** 4

**Summary:**

The paper introduces ML4MILP, a benchmark of mixed inter linear programs (MILP)s containing 100.000 instances across more than 70 dataset classes defined via structure and embedding similarity metric based on bipartite graph representations.
ML4MILP tries to fill the gap of having a large scale, standardized benchmark for evaluating mixed inter linear program solvers.
The authors demonstrate utility of the benchmark by comparing traditional exact solvers and heuristic algorithms and ML-based approaches on their benchmark.

**Strengths:**

The paper appears to present an original benchmark collection targeted to the MILP domain with relevance for the ML and DL community due to ML-based approaches becoming popular in this field.
The paper in general but especially the introduction and related work are written well and give a nice self-contained overview of existing solvers and techniques as well as existing related benchmarks and generally the lack of a larger standardized benchmark for MILPs.
Based on structural and embedding distance between instances of benchmarks, the authors show that existing benchmarks often are more heterogeneous than the MILP benchmark they propose which allows for some insight with respect to the structure of the proposed and existing benchmarks.

**Weaknesses:**

While I can understand the motivation for having homogeneous instances within a class, heterogeneity also might be beneficial to illustrate generalization performance of solvers. I currently do not see this topic being much discussed in the paper.

Analyses of benchmark experiments is limited and only performed qualitatively based on raw numbers and visualizations. Maybe the authors could consider adding some statistical analyses on top of hypotheses so that they can illustrate an empirical workflow on the benchmark?

The presentation of the benchmarking results in Section 4.3 could be improved and more detailed. See also questions below.

The overall contribution (introduction of the benchmark and showing the lack of homogeneity of existing classes of instances of benchmarks based on similarity evaluation metrics with respect to structure and neural embedding distances) is somewhat limited in scientific insight and currently results in me having a hard time to vote for acceptance at a main ML and DL conference such as ICLR. Maybe submitting the paper to a Benchmark and Dataset track of a suitable conference might be more targeted?

**Questions:**

In L269 you mention that you "carefully gathered a substantial number of MILP instances through various means". The word "carefully" is used a lot in context with the selection of instances but it is not really explained, how the selection process was performed. Which exactly were criteria for selection instances "carefully"?

Figure 3 shows also anytime performance curves. Is there any reason why such curves were not presented for some benchmarks in the main paper or appendix?

Having a huge benchmark of many instances may risk that users select a subset of instances to benchmark on to reduce computational burden. Do the authors officially recommend to benchmark algorithms always on all instances?

In Section 4.2 which optimizer and loss function was used for the GNN?

Table 2 and 3 appear to show somewhat redundant information, e.g., one shows the objective function values and the other the gap in optimality. Maybe one table could be moved to the appendix to free up some space for additional visualizations of results or statistical analysis of results?

Figure 8 is quite difficult to digest due to many information but also color gradients being used that however represent factor levels (i.e., the ML4MILP benchmark instances). Maybe the authors could find some easier way to visualize this, i.e., separate colors for IP and MILP or split the figure and use qualitatively different colors instead of a color gradient?

In Appendix D.2 you state:
D.2 "we appreciate the reviewers attention to this detail"
This makes me wonder if this is a re-submission without spending time to clean up the incorporation of previous feedback.
Especially as the section in the main paper concerned with using the standard normal density function on benchmark scores (gap estimates) seems to not have been improved and lacks details and clarification why and how exactly this transformation was performed.
Also, I noted that some descriptions are not really clear here with respect to this transformation.
While the CLT is concerned with the limiting distribution of the sample mean, this is not really applicable here or justifies using the density function of a normal distribution directly to transform the gap scores.
Maybe the authors can improve this section (starting from L509 in the main paper) and explain in detail how and why this transformation was performed?
E.g. what was the reason to not perform a min-max or quantile normalization?
Also, when using the density function of a normal, how is it guaranteed that the transformed scores will be on a range from 0 to 100?

L951 easy - medium - hard <-> tens of thousand, hundreds of thousand and millions of decision variables.
Is this the standard way to classify problems in the related literature or can the authors justify this?

I, personally, found it unusual, that Deepmind is explicitly mentioned two times in the related work section within 20 lines.
Surely, it is important to show that research groups are working on this topic but explicitly mentioning one group of one big tech company does in my opinion not fit the usual style of a scientific paper.
Simply citing the authors, in my opinion, should be enough.

For me, downloading the code from https://anonymous.4open.science does no longer work.
Could it be that the download ability must be refreshed or some functionality is expired?
(The link itself is up but downloading the repository always fails for me.)

**Details Of Ethics Concerns:**

There are no ethical issues.

---

> ### Comment · Reviewer_sUsa · 2024-11-26
>
> As the authors have not provided a rebuttal, I will keep my score.

---

### Official Review · Reviewer_BY5n · 2024-11-04

**Soundness:** 2
**Presentation:** 1
**Contribution:** 1
**Rating:** 3
**Confidence:** 4

**Summary:**

This paper focuses on developing a benchmark ML4MILP to evaluate machine learning (ML) based methods for solving mixed-integer linear programs of MILPs. The benchmark includes MILP instances from various existing MILP benchmark suites such as MIPlib and previous competitions such as ML4CO, as well as synthetically generated large scale instances from canonical MILP instances. For each set of problem instances, the authors obtain both a structural embedding as well as a neural embedding for each problem instance, where the neural embeddings are obtained by representing the MILP instances as bipartite graphs, and leveraging a graph autoencoder. These embeddings are utilized to generate inter-class distances and intra-class distances. These distances are used to highlight the homegeneity and heterogeneity of various classes of instances, and utilized to split heterogeneous classes in smaller homogeneous classes. Given these classes of MILP instances, various off-the-shelf MILP solvers (such as Gurobi) and ML based techniques (such as Predict+Search) are evaluated for a given runtime threshold, and their objective values and gap estimates are presented.

**Strengths:**

Standardized benchmarks are often very useful in the development of any particular area, so a benchmark for evaluating ML based schemes for solving MILPs can be very useful for the area.

**Weaknesses:**

The main weakness of this paper is the proper motivation and description of various choices in the benchmark structure. Specifically:
- (W1) It is not clear why homogeneity within classes is necessary for evaluation of ML based schemes
  - (W1.1) It is not clearly motivated why we need homogeneity? If we are comparing ML based methods to non-ML based methods (such as standard branch and bound), we would want to expose the ML as heterogenous training set as possible to be able to generalize well to unseen problems. Why are we changing / simplifying the problem setup to make sure the ML method works well (as shown in Figure 6 and 7 where the homogenization process makes training and generalization easier)?  Can the authors provide empirical evidence or theoretical justification for why homogeneity is beneficial in this context, and how does homogeneity help or hurt the ability to generalize to heterogeneous problems?
  - (W1.2) To motivate homogeneity, it is also important to clarify what "distances" between problem instances signify. There is no discussion of any intuition as to what it means to have a pair of instances with small or large distance. If two instances are close, does training a ML model on one imply that we can solve the other one fast or to optimality (and if so, how much faster than a non-ML based solver or how much improved optimality)? If two instances are far apart, does that mean a ML model trained on one will not be able to even solve the other one? On a higher level, why is this metric expected to lead to higher homogeneity among problem instances within a "class" subset?  Can the authors discuss the practical significance of these distances in terms of ML model performance and generalization, with any supporting results demonstrating how generalization is related to distances between instances.
  - (W1.3) Furthermore, if there is some notion of distance that implies generalization to unseen problem instances, it is not clearly motivated why the structure based or neural embedding based distances are the right metric.
  - (W1.4) Also, if the distance (whatever is reported in Table 1; it is not clear what the quantity is) is so low, of the order of $10^{-8}$, is there any difference in the embeddings of the instances? This seems like a case where the GNN oversmooths and all instances have effectively the same representation.
  - (W1.5) The "Embedding Distance" in equation (2) needs further clarification. Graph autoencoders allow us to generate a representation for each vertex or edge in the graph (Figure 4 seems to imply just vertices). However, how are these vertex/edge representations used to compute the "Distance" between two different instances $\mathcal{I}_i, \mathcal{I}_j$ since there might be different number of vertices, edges, etc. How are the vertex embeddings converted to a graph embedding?  Can the author describe how the vertex/edge embeddings are aggregated into a single graph embedding? Furthermore, please consider discussing how the choice of aggregation can lead to really small pairwise distances of the order of $10^{-8}$  in W1.4.
  - (W1.6) If I read the appendix correctly, the numbers in Table 1 and Figure 5 are generated from samples from the libraries of MIP problems, and not the complete set of problems. So the "neural embedding distance" is dependent on the set of instances used to train the embedding networks, and thus can vary significantly based on the sample of instances selected (especially if the fraction of samples is very small). It is important to understand how robust this notion of learned embeddings/distances to the data/subsample used for learning (not to mention, the details of the graph autoencoder hyperparamters).
- (W2) It is not clear how the evaluations are set up based on the proposed benchmark, and it is not well motivated why this is the right way to benchmark ML based MILP solvers.
  - (W2.1) The evaluation in Tables 2 and 3 seem to show just one aspect of the setup, thus making the conclusions (such as how ML based schemes still underperform off-the-shelf solvers like Gurobi) somewhat limited in scope. One of the main motivation for introducing ML into the solution of MILP is to be able to solve similar problems quicker or equivalently get to a moderately strong solution fast which can then be used as a seed for off-the-shelf solvers. In that case, an evaluation that considers the time vs objective/gap tradeoff might be more useful as it will highlight the different regimes where ML based schemes would outperform or underperform off-the-shelf solvers. Since off-the-shelf solvers are generic algorithms (not to mention, with decades of research and engineering), they provide guaranteed performance given enough time. It is not clear how the time thresholds were selected for each problem class. In contrast, the mock evaluation shown in Figure 3 (bottom right) where we consider time vs performance tradeoff curves would be more informative than a single metric at some time threshold. The results such as in Tables 2 and 3 do not provide that level of information. Can the authors please include time vs. objective/gap tradeoff curves for each solver and problem class, similar to the mock evaluation in Figure 3. Alternately, can the authors please justify the choice of the specific time thresholds used in Tables 2 & 3, and discuss how these choices can change the conclusions drawn from the evaluations?
  - (W2.2) As with any learning based scheme, the performance will be heavily dependent on the size and quality of the training data, and the similarity or difference between the training and test data. The presented benchmark does not seem to say anything about the amount/quality of data available for training, and how the ML based schemes are trained for the purposes of the evaluation. Are they trained separately for each problem class? How does the size of the problem class (the number of instances available, the size of the individual instances) affect the relative performances of the different ML based schemes? All these factors will tell us more about when ML based schemes might work well, and the benchmark description and the result presentation does not properly address this.

### Additional comments:

- (C1) It is not clear what is being said in section 3.1. On one hand, GNNs are claimed to be lossless encoders, on the other hand, the authors claim that GNNs are unable to model out-of-distribution or larger problems. But some larger models can handle that. The discussion is quite verbose, but it is not clear what are we computing the similiarities of and for what purpose?
- (C2) What does "represents the problem of scale being too large to accept the time to collect training samples" and "band training" in Table 2 caption mean?
- (C3) Table 1 needs more explanation. Are the different rows different "classes" of problems, with the "distances" measuring the pairwise (average?) distances between the instances in the class?
- (C4) Appendix C.3 does not seem to have the mentioned "Detailed classification results". It just states the classes for each MILP library. I did not see any training or validation curves for each of them, or the inter-class/intra-class distances. It would be good to point to the appropriate part of the paper.

**Questions:**

Beyond some of the questions raised in the weakness, here are some additional questions for my own understanding:

- (Q1) How does this benchmark compare and contrast against the benchmark created for the ML4CO Neurips competition (Gasse et al., 2022)? How does this competition benchmark (and the existence of MIPlib) affect the "first open-source benchmark dataset" claim in line 74?
- (Q2) Is the GNN autoencoder learned for each problem "class" or one encoder across all problems? It seems like it is the first, but it is not clear why we should not just consider all instances to train the graph autoencoder.
- (Q3) What is Figure 2 supposed to show? The figure on the left is not a bipartite graph so what is it showing?
- (Q4) It seems like the "Structural Embeddings" discussed in lines 214-215 and 230-232 are manually crafted structural features of any given MILP. Is that correct?
- (Q5) Figure 8 is quite unclear and needs further elaboration. If each color block is a problem, how are bars stacked up? From Table 3, it seems lower gap is better, but then it is not clear whether higher score is better or worse. For the conversion of a particular gap estimate $x$ to a score $\text{Score}(x)$, it is not clear what are the $\mu, \sigma$, and how many samples of scores are generated from the normal distribution.
- (Q6) Why such focus on the graph autoencoder based representations of MILPs? If we want a metric between graphs, can we not employ any graph kernels [A] to find pairwise similarities (which are then extended to compute inter-class and intra-class distances)?
- (Q7) The training curves in Figures 6 & 7 are oddly very drastic for the orange curves (after reclassification). This seems a bit odd. While the x-axis does not have tick labels, it seems like all the learning happens in a single epoch, which is a bit odd. What is it about the set of problem instances that all the learning happens in a single epoch? Is this consistent with existing papers?

[A] Kriege, Nils M., Fredrik D. Johansson, and Christopher Morris. "A survey on graph kernels." Applied Network Science 5 (2020): 1-42.

**Details Of Ethics Concerns:**

There are no ethical concerns in my opinion.

---

### Public Comment · ~Bistra_Dilkina2 · 2024-11-25
**Concerns about Similarity with Distributional MIPLIB and Lack of Citation**

We are deeply disappointed to see that Submission6997 Authors claimed that they proposed the **first** standardized dataset for evaluating ML-based MILP methods, without acknowledging the Distributional MIPLIB dataset, which was introduced in this paper below:

Weimin Huang, Taoan Huang, Aaron M. Ferber, Bistra Dilkina “Distributional MIPLIB: a
Multi-Domain Library for Advancing ML-Guided MILP Methods”. [arXiv link: https://www.arxiv.org/abs/2406.06954v1; Dataset link: https://sites.google.com/usc.edu/distributional-miplib/home ]

The first version of Distributional MIPLIB was submitted to arXiv on June 11, 2024.

TL;DR: Distributional MIPLIB establishes the first comprehensive standardized dataset of benchmarks of MILP distributions for ML-guided MILP solving research; Distributional MIPLIB contributes two real-world MILP domain benchmark instance sets that have never been used in prior research on ML-guided MIP solving research. Surprisingly, Submission6997 claims to also be the first benchmark of ML-guided MIP and contributes the same two new domain datasets.

In the arXiv Distributional MIPLIB paper, we introduced the Distributional MIPLIB, which is a dataset designed for advancing research on ML-guided MILP methods. We have recognized that despite the increasing popularity of ML-for-MILP research, there is a lack of a common repository that provides distributions of similar MILP instances across different domains, at different hardness levels, with standardized test sets. Distributional MIPLIB was designed to address this gap. We curated MILP distributions from existing work in ML-for-MILPs as well as real-world problems and classified them into different hardness levels. Distributions in our work refer to MILPs of the same problem formulation constructed from data parameters sampled from a given distribution, which provide a counterpart for the canonical MIPLIB that consists of heterogeneous instances. We performed experiments to identify challenges and potential areas for improvement in the ML-for-MILP field, which illustrated the benefits of the dataset. Distributional MIPLIB is publicly accessible at  https://sites.google.com/usc.edu/distributional-miplib/home. The website contains details on the sources, mathematical formulation, and instance parameters of the instances and links for downloading the dataset.

Submission6997 did not cite Distributional MIPLIB, despite the similarity in the objective, contributions, and even datasets (down to specific real-world domains chosen for inclusion out of the broad set of possibilities). Instead, Submission6997 Authors claimed multiple times (Lines 74, 80, 88, 144, 160, 532) that their proposed dataset is the first standardized dataset for evaluating ML-guided MILP methods.

Problems generated in Submission6997 seem to closely mirror the selection of MILP domains from Distributional MIPLIB. Submission6997 Authors generated MILPs from nine problem domains as the second source (Line 267) of their dataset, and eight out of these nine domains have already been included in the Distributional MIPLIB paper and website. Six of the eight domains are relatively well known and have been used in prior work on ML-guided MIP research, so one could give the benefit of the doubt that the Submission6997 Authors ‘simply’ converged to the same selection as Distributional MIPLIB. However, for the other two domains in common between the two papers (SNPCC, MMCW), Distributional MIPLIB chose two real-world domains to introduce as novel distributional benchmarks to the ML-guided MIP research community, which 1) were not treated distributionally in their original papers, and 2) our author team had access to the real-world data from these domains, hence enabling truly more structured real-world benchmarks than the other often used domains (largely relying on random graphs). For us, these two domains made sense as 1) we are the authors of the original paper that introduced SNPCC in the context of resilient water infrastructure planning, and 2) MMCW was introduced by authors and collaborators at the NSF AI Institute for Advances in Optimization (ai4opt) which we are part of as well.

Yet, the Submission6997 Authors  **claim these two as their own intellectual contribution to make available in their paper as ‘canonical domains’ as well*** (Line 281). Given that there is a vast variety of real-world domains with MILP encodings in the literature that could be considered for inclusion as distributional benchmarks for the ML-guided MIP community to use, **it is surprising and unlikely that -by chance- **  the Submission6997 authors chose exactly the same 2 real-world problems (especially because they give no specific reason for their choice in Submission6997, such as availability of real data). This points very strongly that the Submission6997 authors indeed heavily borrowed from the intellectual contribution of DIstriobutional MIPLIB.  [end of Part 1]

---

> ### Public Comment · ~Bistra_Dilkina2 · 2024-11-25
> **Concerns about Similarity with Distributional MIPLIB and Lack of Citation [continued]**
>
> The nine problem domains generated by Submission6997 Authors are listed in Lines 281-286, and the ones that overlap with Distributional MIPLIB are:
>
> (1) Combinatorial Auctions
> (2) Set Covering
> (3) Maximum Independent Set
> (4) Minimum Vertex Cover
> (5) Capacitated Facility Location
> (6) Item Placement
> (7) MILP Formulation in this paper: Greening, L. M., Dahan, M., & Erera, A. L. (2023). Lead-time-constrained middle-mile consolidation network design with fixed origins and destinations. Transportation Research Part B: Methodological, 174, 102782.
> We referred to this formulation as Middle-Mile Consolidation Network (MMCN) in Distributional MIPLIB
> Submission6997 Authors refer to this formulation as Middle-mile Consolidation Problem with Waiting Times (MMCW)
> (8) MILP Formulation in this paper: Huang, T., & Dilkina, B. (2020, June). Enhancing seismic resilience of water pipe networks. In Proceedings of the 3rd ACM SIGCAS Conference on Computing and Sustainable Societies (pp. 44-52).
> We referred to this formulation as Seismic-Resilient Pipe Network (SRPN) Planning in Distributional MIPLIB
> Submission6997 Authors refer to this formulation as Steiner Network Problem with Coverage Constraints (SNPCC)
>
> To our knowledge, MMCN/MMCW (7) and SRPN/SNPCC (8) were selected for the first time as benchmarks for evaluating MILP-solving (either ML- or non-ML-based) in Distributional MIPLIB. MMCN and SRPN are generally not considered canonical MILP problems; they are relatively new formulations proposed to address specific real-world applications. To our knowledge, MMCN and SRPN have not been used as standard problems for benchmarking MILP solving (either ML- or non-ML-based) before the introduction of Distributional MIPLIB (and generators or datasets had not been publicly available before Distributional MIPLIB). MMCN and SRPN included Distributional MIPLIB to cover real-world problems in e-commerce and sustainability, as MMCN and SRPN were proposed to address middle-mile network planning problems with lead-time constraints and to enhance the post-disaster resilience of water pipe networks, respectively. Submission6997 Authors did not provide the criteria and reason for selecting these problem domains.
>
> We are **deeply disappointed to see that Submission 6997 did not cite or acknowledge our work**, despite the similarity in motivation, contribution, and dataset. Beyond the problem of not citing the paper, **we want to emphasize the problematic striking similarity between the two papers, unequivocally exemplified by the unlikely choice of the same two real-world domains** (MMCN and SRPN) as new distributional MIP benchmarks.
>
> Bistra, Weimin, Taoan, Aaron

---

> ### Author Response · Authors · 2024-11-25
> **Clarification on this issue**
>
> Dear Bistra, Weimin, Taoan, Aaron, and ICLR Chair,
>
> Thank you for sharing your public comments. We sincerely apologize for any issues and troubles our work may have caused.
>
> First, we would like to clarify that our work was initially submitted to NeurIPS 2024. Given the time of submission (June 5, 2024) and your ArXiv paper online date  (June 11, 2024), I believe it was safe at that time for us to claim a “first” in our submission. You are welcome to verify this with the NeurIPS 2024 chairs.
>
> During the NeurIPS rebuttal process, reviewers recommended including more real-world scenarios. In response, we searched online and came across your work, which inspired us to incorporate two similar scenarios. However, all the data in our paper was generated independently by our team. We sincerely apologize for not citing your work—this oversight occurred due to desynchronization within our author team.
>
> Given the short timeframe between the NeurIPS 2024 decision date and the ICLR 2025 submission deadline, the author team submitted largely the same version of the NeurIPS paper. We will address this in future submissions by revising the paper and ensuring proper citation of your work and other relevant ones.
>
> We also sincerely welcome your technical comments and will carefully consider them during the revision process. Thank you for bringing this matter to our attention.
> Please feel free to let us know if you have any further concerns.
>
> Best regards,
> Authors of 6997

---

> > ### Public Comment · ~Bistra_Dilkina2 · 2024-11-25
> > **further response**
> >
> > Thank you for your response.
> >
> > We believe that continuing to claim a “first” in your ICLR submission - despite your awareness of our Distributional MIPLIB paper -  suggests a failure to give appropriate credit.
> >
> > While you state that your inclusion of MMCN and SRPN was “inspired” by Distributional MIPLIB, including the same real-world domains raises questions about the novelty of your dataset design. In fact, including these domains after seeing their use and treatment in Distributional MIPLIB in relation to ML-guided MIP WITHOUT citation and acknowledgment is a clear violation of intellectual credit.
> >
> > Ignoring these two points (being first and contributing these datasets) in the ICLR submission is an overclaim of contribution, which can mislead reviewers and the greater community.
> >
> > To clarify, Distributional MIPLIB is the **first public** comprehensive standardized dataset for evaluating ML-guided MILP methods. We also submitted our work to NeurIPS 2024 on June 5th, 2024, and you are also welcome to verify this with the NeurIPS 2024 chairs. More importantly, *** we are the first ones to make the datasets publicly accessible (your paper does not link or provide actual datasets that the community can already use)***. We would appreciate it if you provide an updated manuscript that reflects the correct contribution of Distributional MIPLIB and of your work.
> >
> > Additionally, we noticed a discrepancy in your description of the generated SRPN instances in Table 6 of your submission. According to the SRPN problem formulation (Huang & Dilkina, 2020), the number of constraints should be greater than the number of variables, which does not seem to be true according to your manuscript. There may be an error in your statistics or your re-implementation of our models.

---

> ### Author Response · Authors · 2024-11-26
>
> Dear Bistra,
>
> Hi, we sincerely appreciate your follow-up comments and suggestions. We also realized those problems, and will revise the paper accordingly.
>
> Best regards,
> Authors of 6997

---

### Note · Authors · 2025-01-15

I have read and agree with the venue's withdrawal policy on behalf of myself and my co-authors.